# Severe inflammation and lineage skewing are associated with poor engraftment of engineered hematopoietic stem cells in patients with sickle cell disease

In sickle cell disease (SCD), the $\beta6^{Glu \to Val}$ substitution in the β-globin leads to red blood cell sickling. The transplantation of autologous, genetically modified hematopoietic stem and progenitor cells (HSPCs) is a promising treatment option for patients with SCD. We completed a Phase I/II open-label clinical trial (NCT03964792) for patients with SCD using a lentiviral vector (DREPAGLOBE) expressing a potent anti-sickling β-globin. The primary endpoint was to evaluate the short-term safety and secondary endpoints included the efficacy and the long-term safety. We report on the results after 18 to 36 months of follow-up. No drug-related adverse events or signs of clonal hematopoiesis were observed. Despite similar vector copy numbers in the drug product, gene-marking in peripheral blood mononuclear cells and correction of the clinical phenotype varied from one patient to another. Single-cell transcriptome analyses show that in the patients with poor engraftment, the most immature HSCs display an exacerbated inflammatory signature (via IL-1 or TNF-α and interferon signaling pathways). This signature is accompanied by a lineage bias in the HSCs. Our clinical data indicates that the DREPAGLOBE-based gene therapy (GT) is safe. However, its efficacy is variable and probably depends on the number of infused HSCs and intrinsic, engraftment-impairing inflammatory alterations in HSCs. Trial: NCT03964792

Sickle cell disease (SCD) is a highly prevalent autosomal recessive genetic disorder caused by a single point mutation in the HBB gene coding for β-globin. The consequent β6Glu→Val substitution leads to sickle hemoglobin (HbS) polymerization and red blood cell (RBC) sickling under hypoxic conditions, which then results in chronic hemolytic anemia and multi-organ damage. The transplantation of autologous, genetically modified hematopoietic stem and progenitor cells (HSPCs) is a promising treatment option for patients lacking a human leukocyte antigen (HLA)-identical sibling donor. A β-globin-expressing lentiviral vector (LV) called GLOBE[1,2] has been used in a clinical trial in patients affected by β-thalassemia[3]. We recently adapted the GLOBE vector for use in SCD gene therapy (GT) by introducing three polymerization-inhibiting amino acid substitutions into the β-globin chain (βAS3) (ref. 4, DREPAGLOBE vector). It is noteworthy that the BB305 vector currently used in clinical trials[5–8] expresses a therapeutic globin containing only one of the three above-mentioned antisickling amino acids. The DREPAGLOBE vector efficiently transduced SCD bone marrow (BM) HSPCs. Expression of the anti-sickling β-globin reduced the incorporation of the βS-chain into hemoglobin tetramers, decreased hemoglobin's propensity to polymerize and thus reduced the frequency of RBC sickling under hypoxic conditions by up to 50%[9].

✉ e-mail: m.cavazzana@aphp.fr

In GT for SCD, changes in the BM environment (e.g., inflammation, oxidative stress, and ineffective erythropoiesis) and in the properties of RBCs hamper the harvesting and immunoselection of HSPCs from the BM. This can impact the quality of HSPCs and, as a consequence, the efficacy of GT trials. In fact, chronic inflammation can alter hematopoietic stem cell (HSC) engraftment and can bias the HSCs toward the myeloid or megakaryocytic lineages[10–12].

We recently demonstrated that plerixafor can be safely used to mobilize HSPCs in patients with SCD. Although plerixafor-mobilized cells from SCD patients (like BM HSPCs) are characterized by the expression of inflammatory genes[13], our preclinical studies showed that the cells can be efficiently transduced with the DREPAGLOBE vector and can engraft in immunodeficient mice.

Therefore, a Phase I/II open-label clinical trial was initiated in November 2019 (NCT03964792; EUDRACT 2018-001968-33) in Necker Hospital (Paris, France). The trial's objective was to evaluate the safety and efficacy of GT of SCD by transplantation of autologous HSPCs transduced ex vivo with the DREPAGLOBE LV expressing the βAS3 globin gene. Here, we report on the trial's clinical outcomes and consider the factors that can influence (i) the engraftment of genetically modified cells in patients with SCD and thus (ii) the efficacy of GT of this devastating disease.

## Results

### Patient disposition and transplantation

From November 2019 onwards, six homozygous patients with SCD were included in the NCT03964792 GT trial (first patient on November 11, 2019 and last patient on January 1, 2022; Fig. 1a). Patient (P)1 harbored a single 3.7-kb α-globin gene deletion and P3 harbored the rs7482144 single nucleotide polymorphism (SNP, typically associated with elevated fetal Hb levels) in both *HBG2* fetal gamma-globin promoters[14] (Supplementary Table 1). One patient withdrew from the trial because of persistent hemolytic anemia and the last one was not

treated after inclusion because of the unsatisfying results in the previous treated patients. Given the evidence of clonal hematopoiesis[14] and an elevated incidence of leukemia[15] in patients with SCD, and in addition to the standard inclusion criteria for GT trials, we decided to include only patients with no evidence of clonal hematopoiesis. To this end, we used next-generation sequencing (NGS) to analyze a panel of genes previously described as being involved in the emergence of clonal hematopoiesis using genomic DNA from peripheral blood mononuclear cells (PBMCs) and plerixafor-mobilized HSPCs. Although P1, P2, P3, and P4 did not have any mutations with a variant allele frequency (VAF) greater than 1%, while one pre-screened patient was excluded after we detected suspected clonal hematopoiesis in plerixafor-mobilized HSPCs (a *RUNX1* mutation; VAF: 17%). It is noteworthy that P2 was one of the three patients included in the NCT02212535 trial, which demonstrated the safety and efficacy of plerixafor HSPC mobilization[13]; the other two patients in the NCT02212535 trial refused to participate in the GT trial.

Prior to GT, all the patients suffered from severe SCD, which was not controlled by strict adherence to supportive treatment (Table 1). In particular, the patients experienced frequent grade 3 vaso-occlusive crises (VOCs) and episodes of life-threatening acute chest syndrome (ACSs) that were not controlled by RBC exchange transfusions and/or hydroxyurea (HU) treatment (Table 1).

The patients' HSPCs were mobilized with plerixafor and then transduced using our optimized protocol (Supplementary Fig. 1 and Supplementary Table 1); the mean ± standard deviation VCN was 0.9 ± 0.2 in liquid cultures (Supplementary Table 1). Before transplantation, the patients were hypertransfused so that their HbS levels did not exceed 30%. At the time of the apheresis, HbS levels were similar between patients, with no particular difference in the hemolysis markers (Supplementary Table 2). After busulfan-based myeloablative conditioning, the patients received the DP at a dose of 5.97 to 8.1 × 10^6 per kg (Supplementary Table 1). Flow cytometry analysis showed that

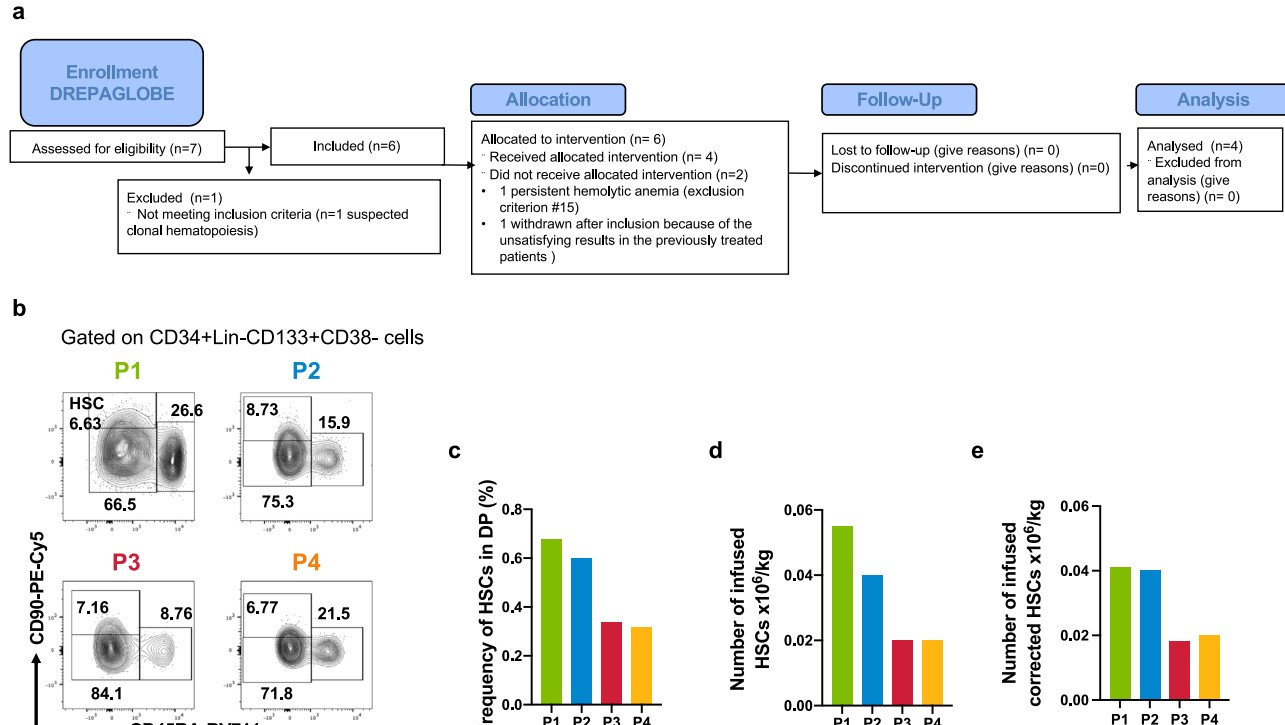

**Fig. 1 | The NCT03964792 phase I/II Open Label Study and the characterization of the DP. a** Flow diagram of the DREPAGLOBE study (NCT03964792). **b** Gating strategy to identify HSCs using flow cytometry. We plotted the frequency of HSCs in the DP (**c**) the number of infused HSCs/kg (**d**) and the number of infused corrected HSCs/kg (**e**). Source data are provided as a Source Data file.

Article

**Table 1 | Clinical evolution before and after GT**

| Patient (age at GT time, y) | | P1 (19) | P2 (29) | P3 (34) | P4 (15) |
|---|---|---|---|---|---|
| **Events before GT** | Grade 3 VOC (N/y) | 14 (0.7) | 30 (1) | 27 (0.7) | 21 (1.3) |
| | ACS (n) | 6 | 2 | 5 | 4 |
| | Concomitant disorders (except iron overload) | Osteomyelitis HBP PRESS sdr Guillain Barré sdr | Hearth hypertrophy Nephropathy | Hypertension Endocarditis Osteonecrosis Nephropathy | Scoliosis |
| **Treatments before GT** | ET (period-y) | 14 | 16 | 5 | 5 |
| | RBC Transfusion Units (N Units per y) | 385U (35 U/y) | 365U (19.21 U/y) | 445 U (22.25 U/y) | 125U (9.6U/y) |
| | HU (period-y) | 3 | 13 | 10 | 6 |
| **Events after GT** | Grade 3 VOC (delay of the first episode from GT (M))(N/year) | 3 (5.6 m) (1) | 2 (8.3 m) (0.6) | 1 (2.8 m) (0.5) | 3 (2.23 m) (1.5) |
| | Mean hospitalization length (days) | 3 | 4 | 71 | 8 |
| | ACS (n) | None | None | None | None |
| | SAE (not SCD linked) | Febrile neutropenia° | None | Neuroendocrine tumor | Mucositis[a] |
| **Treatments after GT** | ET (M post-GT) | None | None | 2.9[b] | 10.3[c] |
| | RBC Transfusion Units (N Units per y)[e] | None | None | 53 U (26.5 U/y) | 9 U (4.5U/y) |
| | HU (M post-GT) | 7.3[d] | None | 8.9 | 5 |
| | Crizanlizumab (M post-GT)[f] | 14.7 | None | None | None |
| | L-glutamine (M post-GT) | None | None | 9.8 | None |

*GT* gene therapy, *y* years, *VOC* vaso-occlusive crises, *ACS* acute chest syndrome, *HBP* high blood pressure, *ET* exchange transfusion, *HU* hydroxyurea, *M* months, *sdr* syndrome.
[a]After conditioning during aplasia period.
[b]Monthly; stopped since 1 year.
[c]Before potential CVO trigger (Travel to Africa: 2 times).
[d]P1 received HU from month 7 to month 16 at a dose of 15 mg/kg/d. He restarted HU treatment at month 27 at a dose of 20 mg/kg/d.
[e]After discharged.
[f]STOP after 27 months of treatment (EMA recommends revocation of authorization of this treatment in SCD due to poor results of the phase III trial).

the proportions and counts of HSCs in the DP were lower in P3 and P4 than in P1 and P2 (Fig. 1b−e and Supplementary Data 1). Neutrophil and platelet engraftment occurred respectively 14 to 19 days and 19 to 63 days post-GT, except in P3; the latter's platelet count never went below 50,000/mm³ (Supplementary Tables 1 and 3). The delayed platelet engraftment, observed also in other autologous HSC transplantation[16] or GT trials[17], was not related to the number of CD34+ cells transplanted[16] or to iron overload[18]. Importantly, no serious adverse event of bleeding was observed.

## Primary outcomes: safety

No drug-related non-serious adverse events (AEs) or serious AEs (SAEs) were observed (Table 1 and Supplementary Fig. 2a). With regard to non-drug-related AEs and other than the AEs related to SCD, P1 and P4 experienced the transient events commonly observed after autologous transplantation. P3 presented severe abdominal pain at month 2, which led to the diagnosis of a gastrointestinal neuroendocrine tumor (Supplementary Table 3). Further investigations demonstrated that the tumor had been present before GT and was negative for vector integration. After surgical resection, clinical remission was achieved. Importantly, no signs of clonal hematopoiesis (i.e., a mutation with a VAF > 1%) were detected in any of the four patients post-GT, as judged from the results of NGS of PBMCs and BM CD34+ cells. Longitudinal monitoring of the vector integration sites revealed polyclonal reconstitution with no clonal expansion (as shown by a high Shannon diversity index[19]; Supplementary Fig. 2b) and a mean number of unique integration sites in PBMCs of 2247 (range: 1665–3784) for the four patients at the last time point (Supplementary Fig. 2c). The mean population size of active clones involved in hematopoietic reconstitution was estimated to be 28,930 (range: 15,859–52,722) for the four patients 2 years after GT (Supplementary Fig. 2d).

## Secondary outcomes: gene marking and hemoglobin expression

Despite the similar VCN in the DP, gene marking in neutrophils was variable and ranged from 0.2 to 0.9 at the last follow-up (0.4 in P1, 0.9 in P2, and 0.2 in P3 and P4; Fig. 2a and Supplementary Table 1). The drop in VCN was greater in P3 (3.6-fold) and P4 (5.5-fold) than in P1 (1.7-fold) and P2 (1.2-fold) (Supplementary Table 1). The VCN was lower in T cells than in neutrophils because lymphodepleting agents were not used in the GT (Supplementary Fig. 3a). The VCN in BFU-Es and CFU-GMs was similar to the value in neutrophils (Supplementary Fig. 3b).

In P1 and P2 (who no longer required RBC transfusions), HbAS3 levels were stable from month 5 onwards (no donor HbA was detected) and were correlated with the VCN (25 g/L for P1 and 33 g/L for P2 at the last visit; Fig. 2b, c). In P3 and P4, who remained transfusion-dependent, HbAS3 levels were <10 g/L−associated with the low VCN. HbF levels were moderate in all the patients other than P3 after the initiation of HU treatment (see below) (Fig. 2d and Supplementary Fig. 3c, d). P3 responded particularly well to HU treatment; this was probably due to the presence of the rs7482144 SNP and possibly to the conditioning regimen.

In P1, P2, and P3, HbAS3 levels were substantially higher in mature RBCs than in CD71+ immature reticulocytes; this indicates that βAS3 globin-expressing cells had a survival advantage at the last stage of maturation (Supplementary Fig. 3e). After GT, the frequency of HbAS3+ circulating RBCs, the proportion of HbAS3 in HbAS3+ RBCs, and the mean amount of HbAS3 per HbAS3-positive RBC were higher in P1 and P2 than in P3 and P4 and were correlated with the VCN (Supplementary Data 2 and Fig. 2e−h). These results suggest that P1 and P2 exceeded the hemoglobin threshold that potentially enables amelioration of the sickling phenotype. The percentage of HbAS3+ reticulocytes was higher in P1 and P2 than in P3 and P4 but was low overall compared with mature RBCs, and the mean amount of HbAS3 per HbAS3-positive reticulocyte tended to be higher for P2 only; this

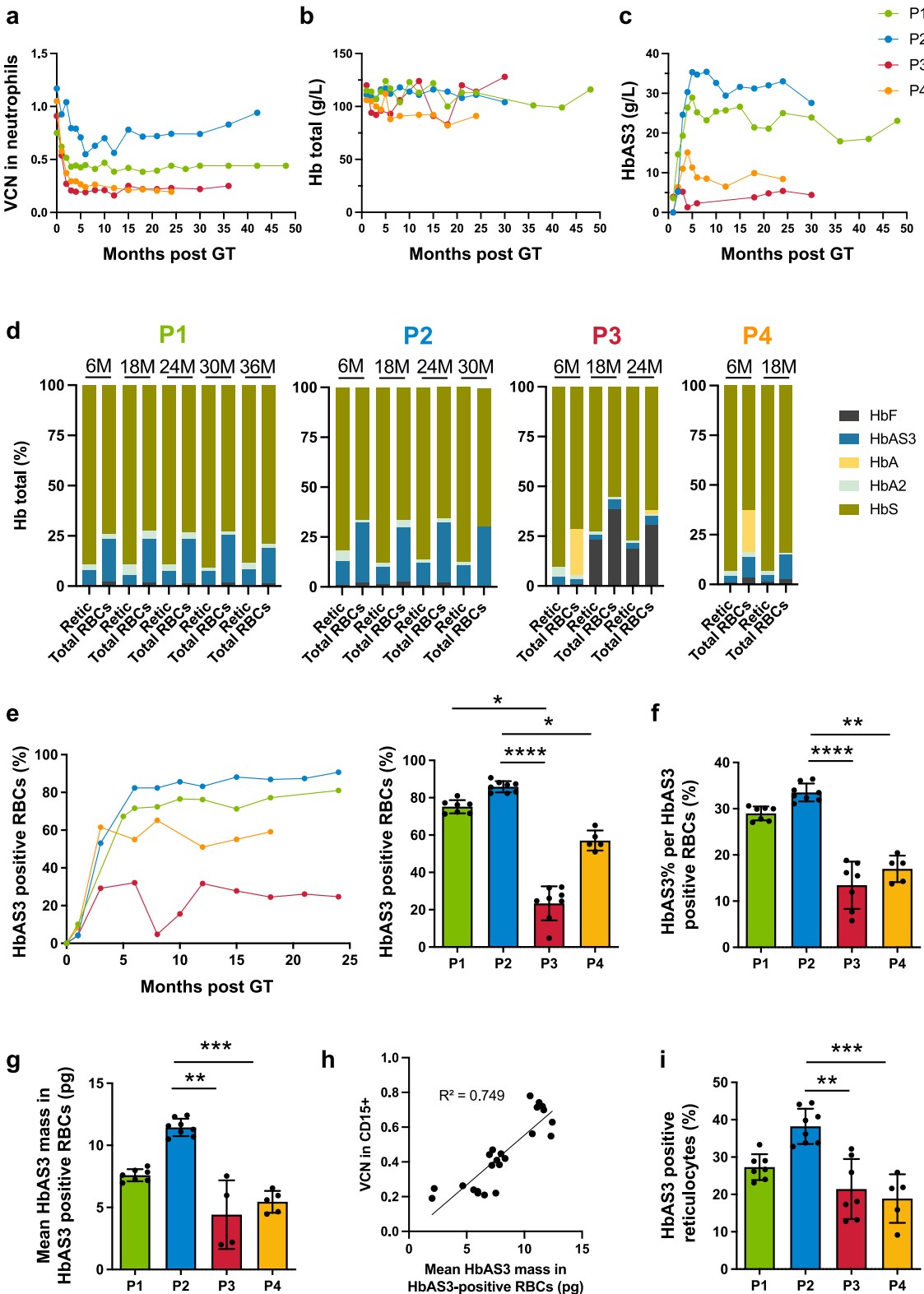

observation further suggests that mostly mature HbAS3-expressing cells had a survival advantage (Fig. 2i and Supplementary Fig. 3e).

## Long-term clinical outcome

The patients' clinical and laboratory variables were followed up in accordance with the study protocol. At the time of the analysis presented here (cut-off date: October 19th, 2023), the median follow-up period was 30 months (range: 18–36 months). Of the four treated patients, two (P1 and P2) had benefited clinically from the GT, becoming transfusion-independent (Table 1). None of the four treated patients presented episodes of ACS, despite the discontinuation of RBC exchange transfusions (for P1 and P2) or the substantial reduction in the frequency of these transfusions (for P3 and P4). P1 and P2 experienced occasional grade 3 VOCs post-GT (mainly triggered by

**Fig. 2 | Gene marking and hemoglobin expression. a** VCN kinetics in neutrophils. **b** Hb concentrations in patients' blood. **c** Kinetics of HbAS3, as assessed by CE-HPLC analysis. **d** Proportions of Hb species (as assessed by CE-HPLC) in reticulocytes vs. RBCs. M, months after GT. **e** Proportion of HbAS3+ circulating RBCs measured by flow cytometry after intracellular co-staining using specific fluorescent monoclonal antibodies directed against HbS and HbA (the latter antibody also recognizes HbA2, which, however, is expressed at very low levels, and HbAS3). Results were obtained by considering only the HbS+ sub-populations to exclude RBCs from transfusion. The *p*-value for the difference between P1 and P3 is equal to 0.0199, between P2 and P3 is <0.0001 and between P2 and P4 is equal to 0.0239. **f** Mean proportion of HbAS3 per RBC, expressed in percentage of total hemoglobin, in HbAS3+ cells (calculated using the formula: HbAS3%-assessed by HPLC/HbAS3+-RBC%-assessed by flow cytometry × 100). The *p*-value for the difference between P2 and P3 is

inferior to <0.0001 and between P2 and P4 is equal to 0.0031. **g** Mean amount of HbAS3 per RBC, expressed in picograms, in HbAS3+ cells (calculated using the formula: HbAS3%-assessed by HPLC × MCH/HbAS3 + -RBC%-assessed by flow cytometry). The *p*-value between for the difference P2 and P3 is equal to 0.0032 and between P2 and P4 is equal to 0.0009. **h** Correlation between the mean amount of HbAS3 per RBC and the VCN in CD15+ cells. **i** Difference in the proportion of HbAS3+ reticulocytes (CD71+ cells), measured by flow cytometry. Histograms in (**e**–**i**) represent mean ± standard deviation of multiple time-points measured from month 5 post-GT (time at which HbA from RBC transfusions was no longer detectable in P1 and P2). The *p*-value for the difference between P2 and P3 is equal to 0.0026 and between P2 and P4 is equal to 0.0010. *$p < 0.05$, **$p < 0.01$, ***$p < 0.001$, and ****$p < 0.0001$ by Kruskal-Wallis test. Source data are provided as a Source Data file.

specific factors, such as cold seawater bath) with shorter hospitalizations and significantly reduced antalgic treatment compared to pre-GT period. After several VOC episodes, P1 started HU treatment; this was effective once the dose had been increased, and no further hospital admissions were noted (Table 1 and Supplementary Fig. 3d). Overall, in P1 and P2, grade 3 VOC occurred with a maximum delay of 24 months from the GT. Since then both patients have not received opioids. P1 and P2 were transfusion-independent after GT, but they both underwent phlebotomy at months 18 and 8, respectively; the goal was to treat liver iron overload and (in P2 only) decrease the blood viscosity (Table 1 and Supplementary Fig. 3d). Finally, for P1 and P2, we have also observed a positive impact of GT on their quality of life and well-being. Despite having undergone GT, P3, and P4 continued to experience severe VOCs: their hospital stays were longer than for P1 and P2, and the analgesic treatment was more intense. We ascribed this difference to the lower initial dose and more rapid post-infusion decline of gene-modified cells (Fig. 2a). As mentioned above, P3 and P4 are still transfusion-dependent and are currently receiving HU. Furthermore, P3 underwent phlebotomy twice and has been treated with L-glutamine after month 9 (Table 1).

### Exploratory outcomes: correction of ineffective erythropoiesis and RBC variables

We recently evidenced ineffective erythropoiesis in the BM of patients with SCD: the ratio between orthochromatic and polychromatic erythroblasts was ~1, rather than the value of ~2 observed in healthy donors (HDs)[20]. Interestingly, GT rescued ineffective erythropoiesis in P1 and P2, whereas P4 showed abnormal erythroid populations before and after GT (Supplementary Table 3, Supplementary Fig. 4a and Supplementary Data 3). P1 and P2's reticulocyte counts decreased after GT (Supplementary Table 3).

The patients' RBCs were characterized before GT and during follow-up. An in vitro sickling assay under deoxygenated conditions showed a significant reduction in the sickling phenotype for P1 and P2 (Supplementary Fig. 4b). The proportion of sickling cells was similar to that observed in a population of SCD heterozygous carriers 5; this was consistent with the near pancellular distribution of HbAS3 in P1 and P2 (Fig. 2). It is noteworthy that before GT and even under normoxic conditions, P2 showed irreversibly sickled cells; this was not observed 12 months post-GT, using quantitative phase microscopy (Supplementary Fig. 4c).

After GT, the maximum deformability under an osmotic gradient and the overall hydration state was better in P1 and P2 than in a group of untreated SS patients (Supplementary Fig. 4d). The maximum deformability under an oxygen gradient was also higher in P1 and P2 than in SS patients (Supplementary Fig. 4e). P1 and P2's sickling points 18 months post-GT were respectively 11.95 Torr and 19.84 Torr; this compares with a median [interquartile range (IQR)] value of 30.2 (26–36.4) Torr in SS patients; hence, P1 and P2's RBCs showed greater anti-sickling capacity under low-oxygen conditions (Supplementary Fig. 4e).

The percentage of dense RBCs was reduced at most of the time points as compared to before GT (Supplementary Fig. 4f). Lastly, the deoxygenation and reoxygenation curves showed an improvement in the hysteresis that stems from the formation and dissociation of HbS polymers[21] (Supplementary Fig. 4g). The P50 values were 29.25 Torr for P1 and 29.22 Torr for P2, which are in the lower range of values measured in untreated SS patients (median [IQR]) 31.6 [29.4–34.5] Torr (ref. 22, confirming the enhanced anti-sickling capability showed by the measurement of RBC deformability under the oxygen gradient (Supplementary Fig. 4e).

### Exploratory outcomes: laboratory markers

Hemolysis markers were evaluated before and after GT. The decrease in hemolysis was evidenced by a lower level of lactate dehydrogenase and a lower reticulocyte count in P1 and P2 and a higher haptoglobin level in P1 (Supplementary Table 3). In P1 and P2, the total bilirubin level (a marker of extravascular hemolysis) fell after GT (Supplementary Fig. 4h and Supplementary Table 3). The intravascular hemolysis was reduced in P1 and (particularly) P2, as shown by a decrease in plasma levels of hemoglobin and heme (both of which were abnormally high before GT) and an increase in plasma hemopexin (Supplementary Fig. 4h). An indirect antiglobulin test revealed no evidence of immune hemolysis.

P1 and P2 had normal white blood cell counts after GT and the level of C-reactive protein (an inflammation marker) normalized in P1 (Supplementary Fig. 4i and Supplementary Table 3).

Even in the absence of RBC transfusions, P1 and P2's serum ferritin levels were still high after GT (Supplementary Table 3); this was probably due to poor compliance with the iron chelation therapy. However, liver iron overload (as assessed by T2* MRI) decreased in P1 and P2 after GT (Supplementary Fig. 4j). In both patients, heart iron levels were in the normal range before and after GT (Supplementary Fig. 4k).

### Exploratory outcomes: transcriptomic analysis of the SCD HSPCs revealed enrichment in inflammatory and committed progenitor signatures

To evaluate the effects of the SCD pathology on HSPCs, we compared the transcriptomic profiles of plerixafor-mobilized HSPCs from patients with SCD and from HDs at the time of collection (i.e., before transduction). In SCD HSPCs, a bulk RNA-sequencing (RNA-seq) analysis showed the upregulation of genes involved in inflammatory responses (as observed previously[13]) and in cell cycle-related processes (Fig. 3a). Accordingly, gene signatures specific for committed progenitors (i.e., earliest thymic progenitors, monocytes, pre-B cells, and cell cycle-G2M) were enriched in the patients' samples. We also observed, in SCD samples, an enrichment in genes typically overexpressed in active HSC (and downregulated in dormant HSC); hence, SCD samples might contain fewer dormant HSCs and more progenitors (Fig. 3b).

We next used ROMA[23] to quantify the activity of specific sets of genes involved in inflammation and oxidative stress in individual

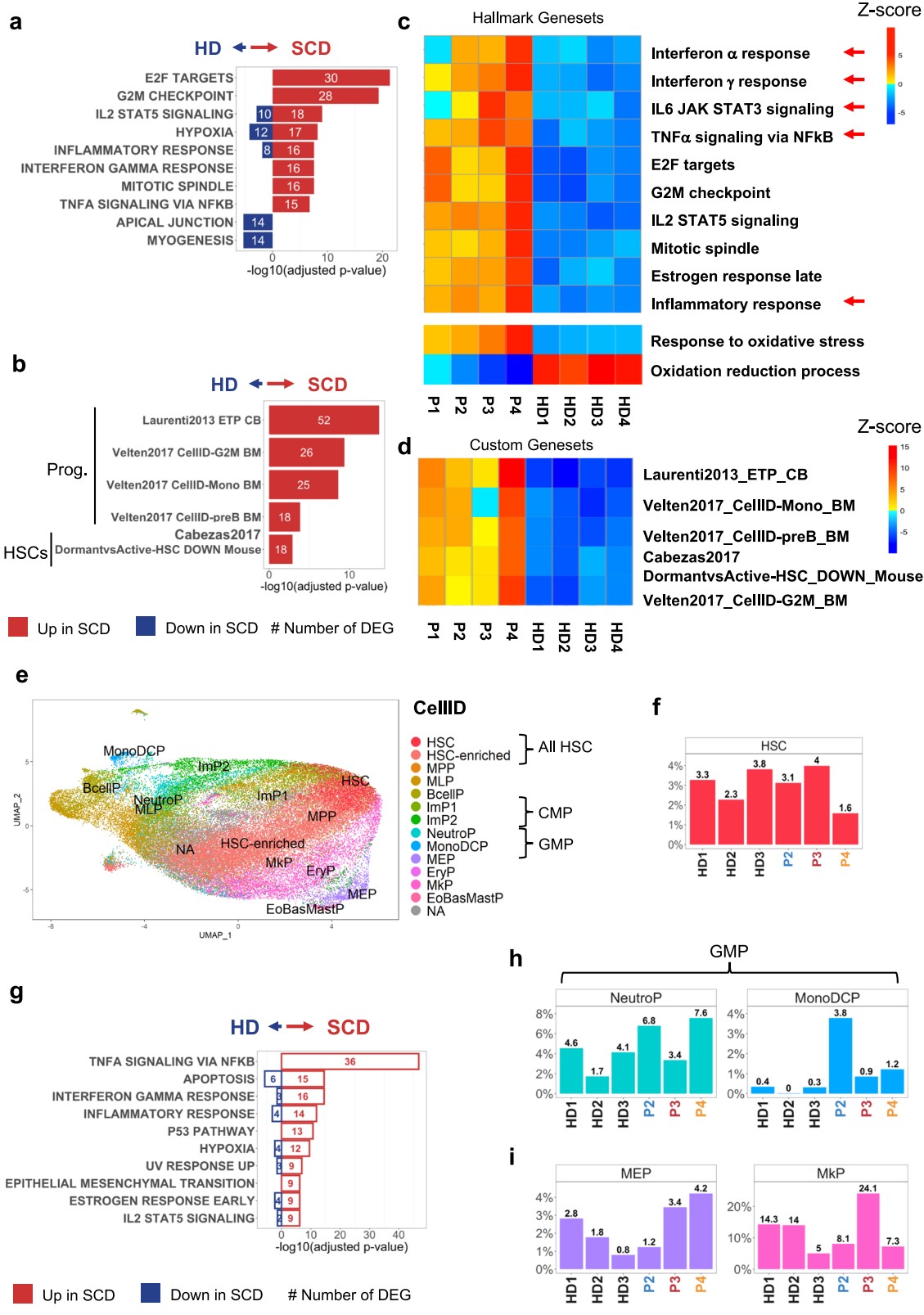

samples. Compared with HDs, patients with SCD had upregulated many different inflammation pathways (Fig. 3c). We observed marked interpatient differences; P3 and P4 (with a defect in long-term engraftment) presented a prominent inflammatory signature. The "Tumor necrosis factor alpha (TNFα) signaling via NFκB" and "IL-6 JAK STAT3" (including IL1β) inflammatory pathways were highly enriched in P3, whereas the genes involved in the interferon (IFN)

alpha and gamma responses were strongly upregulated in P4 (Fig. 3c). Furthermore, P3 and P4's HSPC's strongly expressed genes involved in the response to oxidative stress, and the score for the oxidation-reduction pathway was lower for all the patients with SCD than for HDs (Fig. 3c). Lastly, we observed an increase in progenitor, cell cycle and active HSCs signatures in all patients with SCD (particularly in P4) (Fig. 3d).

**Fig. 3 | Characterization of HSPC composition and gene signatures in SCD patients vs HD cells by bulk and single-cell RNA-seq. a, b** RNA-seq data were analyzed using DESeq2. We performed a two-sided hypergeometric test with MSigDB after correction for multiple hypothesis testing according to Benjamini and Hochberg on the DEG using Hallmark or custom genesets from published transcriptomic data[24,66, 67] (Log$_2$FC > 1.2, FDR < 0.05). **a** Top 10 enriched Hallmark genesets in SCD vs HDs HSPCs. **b** Top 5 enriched custom HSPC genesets in SCD vs HDs HSPCs. Red, upregulated genes in SCD and blue, downregulated genes in SCD in comparison with HDs. **c, d** Representation and quantification of modules of genesets activity with ROMA tool in individual HSPC samples using selected hallmark and gene ontology genesets (**c**) and custom genesets (**d**). Red arrows indicate inflammatory genesets. **e** Unsupervised analysis of 54,689 HSPCs and 15,961 genes from 3 HDs and 3 patients with SCD, represented as two-dimensional UMAP plots. Each individual cell in our dataset was annotated using the Cell-ID method and reference BM HSPC signatures[24]. HSC hematopoietic stem cell, HSC-enriched hematopoietic stem cell-enriched, MPP multipotent progenitors, MLP multipotent lymphoid progenitors, ImP1 & ImP2 immature myeloid progenitors, NeutroP neutrophil progenitors, MonoDCP monocyte and dendritic cell progenitors, BcellP B cell progenitors, MEP megakaryocyte and erythrocyte progenitors, EryP erythroid progenitors, MkP megakaryocyte progenitors, EoBasMastP eosinophil, basophil and mast cell progenitors, NA not annotated. **f** Proportion of the HSCs per individual. **g** Top 10 pathways (in terms of p-value, identified using a two-sided hypergeometric test, MSigDB and hallmark genesets) among the 280 DEGs identified with the MAST tool in SCD HSCs (n = 3) vs. HD HSCs (n = 3). In each pathway, genes that are upregulated in SCD (relative to HDs) are shown in red, and those that are downregulated in SCD are shown in blue. The false discovery rate (−log10(adjusted p-value)) is shown for each pathway. The numbers of upregulated and downregulated genes in each pathway are also shown. **h, i** Bar plots showing the percentages of NeutroP-MonoDCP (**h**) and MEP-MkP (**i**) in each individual.

## Exploratory outcomes: upregulated inflammatory pathways in the most immature HSCs, and an increase in committed progenitors in patients with SCD

To further understand the changes in HSPCs in the context of SCD, we performed single-cell RNA-seq of HSPCs from P2, P3, and P4 and from three HDs[24] (Fig. 3e). Relative to HDs, the proportion of the most immature HSCs was normal in P2 and P3 but slightly low in P4 (Fig. 3f). However, analysis of deregulated genes in this population demonstrated that inflammatory signatures (TNFα signaling via NFκB and IFN gamma response) were enriched in HSCs from patients with SCD (Fig. 3g).

Single-cell RNA-seq revealed an increase in myeloid progenitors in P2 (neutrophil and monocyte/dendritic cell progenitors) and P4 (neutrophil progenitors) (Fig. 3h). This was correlated with high proportions of common myeloid progenitors, granulocyte-monocyte progenitors, and dendritic cell progenitors (identified using a 22-color spectral cytometry panel; Supplementary Fig. 5a, b) and a higher CFU-GM/BFU-E ratio in P2 and P4 (Supplementary Fig. 5c). Flow cytometry also highlighted a high proportion of B cell and natural killer cell progenitors in P4 (Supplementary Fig. 5d), which was in line with the results of the transcriptomic analysis (Supplementary Fig. 6a). Our scRNA-seq (Fig. 3i) and flow cytometry (Supplementary Fig. 5e) analyses revealed a high proportion of megakaryocytic-erythroid progenitors (MEPs) in P3 and P4. Overall, our results showed that patients with SCD have several lineage biases and a high proportion of committed progenitors.

## Exploratory outcomes: megakaryocyte-biased-HSCs and IL1β pathway upregulation in P3

Single-cell RNA-seq revealed a striking phenotype in P3: the proportion of megakaryocytic progenitors (MkPs) was very high (24%, vs. a mean of 11% in HDs) (Fig. 3i). Furthermore, a t-sne representation of the flow cytometry results showed that MEPs from P3 differed from those from the other patients and from HDs (Supplementary Fig. 5a). Relative to MEPs from HDs, MEPs from P3 were highly positive for the erythroid CD71 marker, and 22% expressed the CD41 megakaryocytic marker; this suggests that P3's MEPs were more committed than MEPs from HDs[25] (Supplementary Fig. 5f–h). Importantly, P3's HSCs (defined as CD34+Lin-CD133+CD38-CD45RA-CD90+ cells) had a particular phenotype with CD71-low expression, which was not observed in the other patients or HDs (Supplementary Fig. 5i). To establish whether the high proportion of committed cells was linked to some type of HSC bias, we used the Cell-ID method to identify cells with signatures of multiple cell types (i.e., in addition to their top significant signature that defines the cell label). Interestingly, P3 had a population of 3340 cells presenting both HSC and MkP signatures; this population was substantially smaller in P2, P4, and HDs (range: 130–743 cells) (Fig. 4a, Supplementary Fig. 6e and Supplementary Data 4). The dual signature suggests that P3 had an abnormally high megakaryocyte-biased HSC population. We next used Cell-ID to extract the signature from this pathological cell population in P3 (Supplementary Data 5 and found that expression of the inflammatory cytokine IL-1β (previously associated with a megakaryocytic bias in HSCs[26]) was significantly upregulated in the most immature HSCs, in the HSC-enriched population (containing less immature HSCs), and in the MkPs (Fig. 4b). This mixed-signature population strongly expresses von Willebrand factor (vWF), a marker of megakaryocyte-biased HSCs in mice (Fig. 4b).

## Exploratory outcomes: high TNF-α and IFN pathway scores in P4's HSCs

HSCs from patients with SCD strongly expressed inflammation-related genes involved in TNFα signaling and IFN gamma response (Fig. 3g). By leveraging the Cell-ID ability to compare individual cell signatures with well-defined gene sets, we more precisely examined these two pathways in individual patients and HDs (Fig. 4c–f). The strongest TNFα signaling (Fig. 4c and e) and IFN gamma response (Fig. 4d and f) scores were found in P4, especially in the most immature HSCs. Accordingly, P4's cells showed more upregulated genes belonging to TNFα (including *MAFF*, *NFKBIA*, and *JUNB*; Fig. 4g and Supplementary Fig. 6g) and IFN signaling signatures (e.g., *IFI44L*, *IFITM3*, and *MX1;* Fig. 4h and Supplementary Fig. 6f). The gene encoding the myeloid transcription factor CEBPB (belonging to the TNFα signaling pathway) was strongly expressed in HSCs from P2 and P4 (who also had high proportions of myeloid-committed progenitors; Fig. 3h) but was not expressed in HSCs from P3 and HDs (Fig. 4g and Supplementary Data 6). These results suggest that in P4, activation of TNFα and IFN pathways led to a myeloid bias in HSCs – as observed previously in other inflammatory diseases[11].

## Exploratory outcomes: inflammatory and aging signatures in HSCs are associated with poor engraftment in patients with β-hemoglobinopathies

To evaluate if inflammation affects HSC engraftment in other diseases, we analyzed G-CSF and plerixafor-mobilized HSPCs from a previous cohort of patients with β-thalassemia who received GT using the BB305 vector (n = 4[7]) together with healthy donor controls. Of note, one out of 4 patients (TDT3) showed a low engraftment of transduced HSPCs associated with a low number of infused bona fide HSCs (quantified by multi-parameter flow cytometry) (Supplementary Fig. 7a–c). Bulk RNA-seq experiments showed that mobilized HSPCs from patients with β-thalassemia display an overall inflammatory profile (Supplementary Fig. 7d, e). Interestingly, in TDT3 (showing a poor engraftment), a stronger inflammation signature in bulk HSPCs and in HSCs identified by scRNA-seq was associated with poor engraftment (Supplementary Fig. 7e–i). In particular, TNFα signaling but not IFN gamma response score was significantly increased in TDT3 compared to the other patients or HD controls (Supplementary Fig. 7h–k).

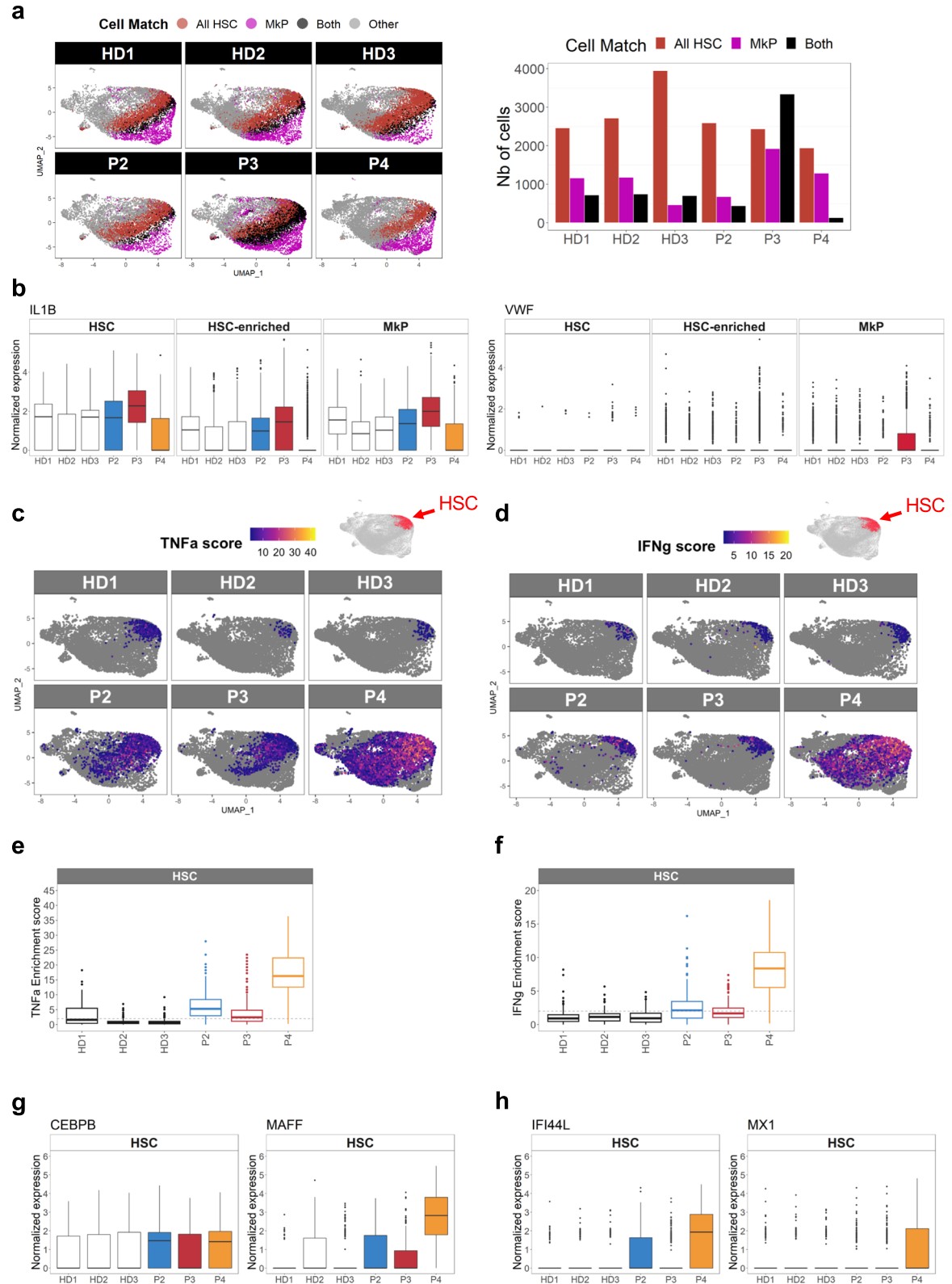

In order to further understand the consequences of inflammation in those SCD and TDT patients, we analyzed a human aging signature[27]. Interestingly, the Aged vs Young HSC signature score was the highest for SCD P3, P4, and TDT P3 in HSCs and HSC-enriched populations (Supplementary Fig. 8a–d). As expected for HDs, the Aged vs Young HSC median score in HSCs was positively correlated with the age ($r = 0.89$; $p = 0.045$). When combining HD with patients, we lost the correlation ($r = 0.38$, not significant), as SCD P3 and P4, and TDT P3

presented with abnormally high Aged vs Young HSC score (Supplementary Fig. 8e).

**Exploratory outcomes: anti-inflammatory ex vivo treatment reduced inflammatory signature in P3 and P4 HSPCs**

We then attempted to correct the SCD inflammatory profile, by treating HSPCs ex vivo from P3 and P4 with JAK inhibitors (ruxolitinib and baricitinib) to target the hyperactivated IFN pathway or with TNF-α

**Fig. 4 | IL1b-driven megakaryocytic-biased HSCs in P3 and TNFa and IFN driven myeloid- bias HSCs in P4.** The Cell-ID method was used to assess the statistical enrichment of individual-cell gene signatures vs. signaling pathway gene sets (such as Hallmark gene sets, MSigDB collections, v7.5.1) based on two-sided hypergeometric test p-values with Benjamini–Hochberg correction for the number of tested gene signatures. Enrichment scores were calculated as the -log10(p-value). **a** UMAP plots highlighting cells significantly matching the All HSC and MkP signatures, for each HD ($n = 3$) and SCD patient ($n = 3$) ($p < 0.05$). Cells matching All HSC and MkP cell types are shown in black, and cells matching with only one cell type are shown in red (All HSC) and pink (MkP). The number of cells in each category and per patient, is depicted in the histogram on the right. **b** Boxplots of *IL1β* and *VWF* mRNA expression in HSCs, HSC-enriched and MkP populations, in each HD ($n = 3$) and SCD patient ($n = 3$). **c, d** UMAP plots of TNFa and IFN gamma response pathway enrichment scores for each HD ($n = 3$) and SCD patient ($n = 3$), determined with Cell-ID. **e, f** Boxplots representing significant TNFa and IFN gamma enrichment scores ($p < 0.01$) in HSCs for each HD ($n = 3$) and SCD patient ($n = 3$). Dotted lines represent the significant threshold $-\log_{10}$(p-value) = 0.01). **g, h** Boxplot representing *CEBPB, MAFF, IFI44L*, and *MX1* mRNA expression in the HSCs populations in each HD ($n = 3$) and SCD patient ($n = 3$). In each boxplot, the edges of the box indicate the first and third quartiles and the center line indicates the data median. The whiskers denote 1.5× interquartile range, data beyond the end of the whiskers are called "outlying" points and are plotted individually.

inhibitors (infliximab and etanercept). After 2 days of culture, untreated SCD HSPCs showed the upregulation of genes involved in inflammatory responses compared to HD HSPCs, as detected by bulk RNA-seq (Fig. 5a). Upon treatment with JAK or TNF-α inhibitors, SCD HSPCs showed a significant reduction in the inflammatory signatures compared to their untreated counterparts (Fig. 5b–d). Importantly, fewer differences were observed between treated SCD HSPCs and untreated HD HSPCs (Fig. 5e–g). Finally, SCD HSPCs ex vivo treatment allowed a significant reduction in the expression of some individual genes involved in inflammatory responses (Fig. 5h) and strongly upregulated in bona fide HSCs (defined by single-cell RNAseq) (Fig. 4h and Supplementary Fig. 6).

## Discussion

Here, for the first time we have featured an LV expressing a potent anti-sickling globin for the treatment of patients with SCD and analyzed the success factors in HSPC-based GT.

In contrast to our starting hypothesis (i.e., that the introduction of two additional anti-sickling amino acids into the therapeutic globin would give better results), the clinical and biological outcomes were similar to those observed with the BB305 vector[7] in patients with comparable VCNs.

Despite this lack of a difference in clinical and biological outcomes, our study is the first to have provided evidence of the influence of the SCD pathophysiology on HSC properties. Two of the four patients showed a partial loss of corrected cells upon engraftment, and this correlated with lineage bias and high levels of inflammatory marks in the HSCs.

The fact that most clinical manifestations of SCD correlate with a high white blood cell count indicates a role for leukocytes and inflammation in the pathophysiology of SCD. Leukocytosis is common in patients with SCD and manifests itself by elevated monocyte and neutrophil counts and high circulating levels of inflammatory cytokines like TNF-α, IL-1, and IL-8[28]. This inflammatory environment in SCD and ineffective erythropoiesis (with erythroid hyperplasia and extensive erythrophagocytosis)[20,29] might significantly reduce the quality, yield and engraftment of HSPCs. We recently highlighted the negative impact of inflammation on HSPCs in chronic granulomatous disease (CGD), a primary immunodeficiency[24]. Similarly, oxidative stress in HSPCs, so far observed only in a mouse model of SCD, might also alter the properties of human HSCs[30,31]. Preliminary research has shown that HSCs from a mouse model of SCD have a reduced engraftment capability compared to HSCs from wild-type mice[30]. Nevertheless, it is still not fully clear how chronic inflammation and oxidative stress affect HSCs in patients with SCD and whether cell manipulation further exacerbates this phenotype.

In the present study, a transcriptomic analysis of bulk HSPC populations from patients with SCD at steady state confirmed the enrichment in inflammatory and oxidative stress signatures—especially in patients showing poor engraftment of transduced cells, and even though the VCN in the DP was similar to that measured in patients with good engraftment. The hypertransfusion regimen that could improve the quality and quantity of collected HSPCs (with lesser dilution by erythroid progenitors) did not prevent HSPCs to display an inflammatory profile in all the patients. However, we cannot exclude that it mitigated it as we have not analyzed HSPCs in patients who are not hypertransfused. Of note, P3 and P4 received fewer RBC transfusion units in the 3 years before inclusion in the study and during the hypertransfusion regimen compared to P1 and P2, because of the lower weight (for P4) or because the patients were responding relatively well to the treatment (for P3) or because a lower number of transfusion units during the hypertransfusion regimen were required to reduce HbS levels below the 30% threshold (for P3 and P4). Therefore, we cannot exclude that the transfusion regimen could have a positive impact on the quality of HSCs.

To assess the impact of this inflammation on the most immature HSCs (those responsible for long-term engraftment, which account for <1% of HSPCs), we performed single-cell transcriptomic analyses.

P3 (with poor engraftment of corrected cells) had an elevated frequency of MkPs and an abnormal population of cells with a mixed HSC/MkP signature. This population strongly expressed IL-1β and vWF and was reminiscent of the platelet-biased vWF+ HSCs detected in murine models at steady state[32] and in the context of acute inflammation, where it contributes to emergency megakaryopoiesis[12]. It has been shown that platelet bias increases with age in mice[33,34] and in humans[35], and that this process is driven by IL-1β signaling[26,36]. Furthermore, we found that P3 had an abnormal erythrocytic/megakaryocytic population expressing CD71 and CD41 and HSCs expressing CD71. The latter result is reminiscent of the erythrocyte/megakaryocyte-biased CD71 + HSC-like population (with no self-renewal capability) reported in extramedullary reservoirs of HSPCs in humans[37]. These results suggest the presence of erythrocyte/megakaryocyte-biased HSCs with poor repopulation ability that could explain the loss of engraftment capacity in P3.

The single-cell transcriptomic analysis in P4 (which showed poor engraftment of corrected cells and had a high percentage of myeloid progenitor cells) evidenced a high inflammation score for the TNF-α and IFN pathways. A higher TNF-α level might contribute to myeloid bias and defective HSC engraftment. TNF-α has been proposed to promote HSC survival during acute inflammation (by activating the NFκB pathway and initiating emergency myelopoiesis). In contrast, it can contribute to HSC aging in the context of chronic inflammation[38]. The enrichment in IFN-alpha/gamma signature, which is associated with the expression of the myeloid CEBPB factor in P4's HSCs, closely resembles the changes that we have described previously in patients with CGD in a very severe, chronic inflammatory context[24]. Of note, similarly to P4, P2 HSPCs showed a myeloid bias. However, despite this similarity, P4 presented with a much higher inflammatory score compared with P2, suggesting chronic accumulation of inflammatory stress that could explain the loss of engraftment capacity in P4.

In conclusion, IL-1β, TNF-α and IFN-alpha/gamma inflammatory signatures were more prominent in SCD HSCs with defective engraftment and correlated with a recently published human aging signature[27]. Chronic inflammation can lead to premature aging of

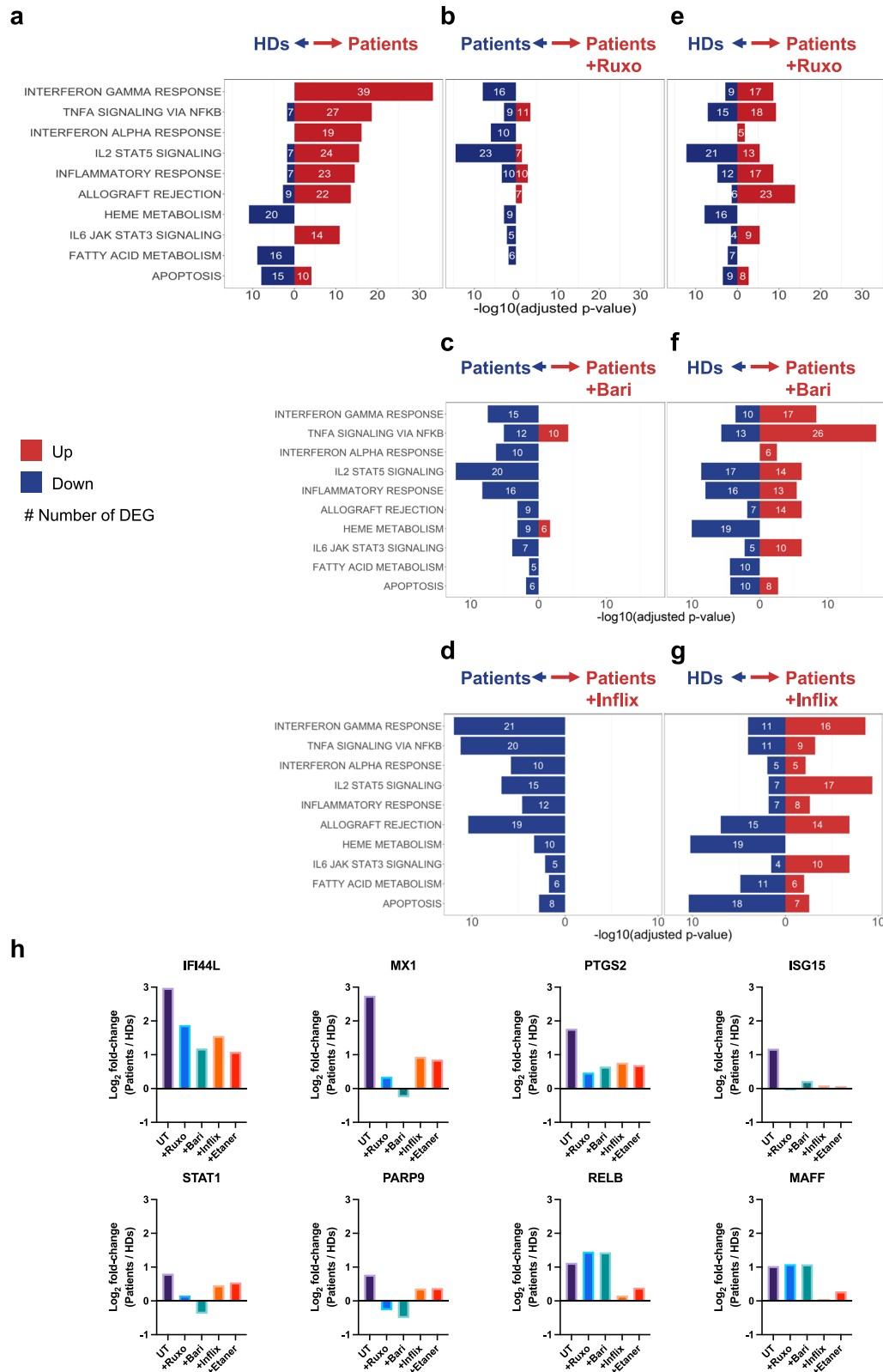

**Fig. 5 | Anti-inflammatory treatment reduces inflammation in P3 and P4 HSPCs.** Transcriptomic analysis of HSPCs from P3 and P4 (untreated or treated with JAK inhibitors: Ruxolitinib, Ruxo, or Baricitinib, Bari or TNF inhibitors: Infliximab, Inflix or Etanercept, Etaner), and HDs ($n = 2$) using two-sided hypergeometric test with MSigDB after correction for multiple hypothesis testing according to Benjamini and Hochberg on the DEG using Hallmark genesets (Log$_2$FC > 1.2, FDR < 0.05). **a** Top 10 enriched Hallmark genesets in untreated patients vs HDs HSPCs cultured for 48 h. **b–d** Enriched genesets (identified in **a**) in untreated vs treated patients HSPCs (**b**, Ruxo, **c**, Bari, **d**, Inflix). **e–g** Enriched genesets (identified in **a**) in treated patients HSPCs (**e**, Ruxo, **f**, Bari, **g**, Inflix) vs untreated HDs HSPCs. **h** Histogram representation of the Log$_2$FC of 8 inflammatory dysregulated genes (Patients vs HDs). UT: Log$_2$FC in untreated patients' HSPCs vs HDs HSPCs. +Ruxo /+Bari /+Inflix /+Etaner: Log$_2$FC in treated patients' HSPCs vs untreated HDs HSPCs. Source data are provided as a Source Data file.

HSCs, which are biased towards the megakaryocytic or myeloid lineages[33,34,39].

The pathways found to be dysregulated in our study are potential pharmacological targets for correction of the SCD HSC phenotype. For example, one could consider administering specific anti-inflammatory treatments to patients before HSPC harvesting or optimizing the ex vivo HSPC engineering process with anti-inflammatory drugs, antioxidants, or compounds that sustain stemness. By way of example, a recent study showed that targeting IL-1 signaling prevented HSC aging in a context of chronic inflammation and might make GT of SCD more effective[36]. The use of anakinra (an IL-1 receptor antagonist approved by the US Food and Drug Administration) could therefore be considered[40]. Alternatively, canakinumab (a monoclonal antibody against IL-1β) has already been used to safely reduce systemic inflammation in patients with SCD[41]. Lastly, it might be possible to use JAK inhibitors to target the hyperactivated IFN pathway or TNF-α inhibitors. In fact, our experiments showed that treatment with these drugs reduced the inflammation signature in P3 and P4. Interestingly, two studies recently shown that ruxolitinib treatment improves self-renewal capacity of HSCs and enhance their maintenance in culture[42,43]. These results pave the way for the development of GT protocols based on the pre-treatment of patients' HSPCs ex vivo or, even better, on the administration of anti-inflammatory drugs to the patients in vivo prior GT to improve the quality of HSCs. To validate the use of these drugs to improve SCD GT, further in vivo studies evaluating the impact on SCD HSCs and the safety profile are required.

The identification of predictive biomarkers would allow HSPCs to be screened using targeted RNA sequencing. Future studies on bone marrow samples of a larger cohort of SCD are ongoing to correlate the cytokine profile (that could also potentially serve as biomarker) in the marrow with the inflammation signature in HSCs. These biomarkers could allow the stratification of patients with a low inflammatory signature score (who could be treated safely with GT) from those with a high inflammatory signature score (who could be pretreated with anti-inflammatory drugs or undergo haploidentical HSC transplantation). In fact, in light of the recent advances in the allogeneic HSC transplantation protocols[44–46] and of the results obtained in our study, the choice of the GT therapeutic option should be carefully evaluated, e.g., favoring haploidentical HSC transplantation when SCD patients' HSCs show a high inflammatory signature score. Nevertheless, as haploidentical HSC transplantation is limited to very experienced transplantation centers in high-income countries as it is associated with higher morbidities and mortality rates compared to GT, it is necessary to continuously improve the GT approach with the aim to make it more feasible and affordable even in non-experienced centers.

Other factors that might have influenced the outcome of GT in our patients with SCD include the number of infused, transduced HSCs, which was lower in patients with graft failure. This aspect could be taken into account in subsequent clinical trials; if the number of HSCs (as determined in our multiparameter flow cytometry analysis) is below a certain threshold, HSC mobilization and collection should be repeated.

It is also possible that non-cell autonomous factors influence HSC engraftment in patients with SCD; for example, inflammation can negatively affect the various components of the BM microenvironment[30,31]. Future analyses of bone marrow biopsies could further shed light on SCD pathophysiology and on the BM inflammatory environment, which could influence the success of both allogeneic and autologous HSC transplantation.

Furthermore, the use of busulfan for myeloablation (which is known to lead to endothelial and epithelial toxicity[47,48]) can potentially worsen the inflammation, and impact on the bone marrow niche and, as a consequence, the GT outcome. Therefore alternative, less toxic options are currently being evaluated, such as a non-genotoxic myeloablative regimen with antibodies anti c-kit to deplete endogenous HSCs[49–51]. Unfortunately, this approach is not currently available for clinical use.

Finally, optimizing the transduction protocol and LV design could also improve the outcome of SCD GT. To ameliorate the outcome of the future clinical trial, we will employ an optimized protocol based on the use of effective transduction enhancers[8,52,53] and an optimized vector that corrects the SCD phenotype with greater potency and hence lower VCN[54].

Maximizing engraftment of genetically modified HSCs is a key goal in SCD—not only to obtain clinical benefit but also to reduce the likelihood of myeloid leukemia reportedly associated with engraftment failure or the use of a conditioning regimen[55]. HSPCs from patients with SCD may have a high mutation burden[31,56] because of chronic inflammation and oxidative stress. Some SCD patients in an LV-based GT clinical trial showed an increase in the frequency of driver mutations potentially involved in the emergence of clonal hematopoiesis during GT[56]. Along with LV-based therapies, new genome editing strategies for SCD have recently moved to the clinic. However, these strategies raise some safety concerns because of their potential genotoxicity[57], which might be exacerbated in patients with SCD. Hence, the additional genotoxic risk potentially associated with current and future novel genome-editing GT strategies for SCD will have to be evaluated carefully in preclinical studies.

Limitations of the study include the relatively small number of patients; however, this was due to our decision of stopping the trial given the unsatisfying clinical results obtained for P3 and P4 that prompted us to critically analyze the success factors in GT of SCD.

In fact, the variability in the clinical results observed in the present study emphasizes the need to establish a pre-GT HSC score based on the number and quality of HSCs (evaluated by flow cytometry and transcriptomic analyses), which, as suggested by our present results, are not necessarily correlated with each patient's specific comorbidities. It is noteworthy that this HSC quality score strategy could be applied to other inflammatory disorders and not just SCD to assess and compare the safety of various HSC-based GT approaches.

## Methods
### Study design
This Phase 1/2 non-randomized, open-label study (ClinicalTrials.gov identifier: NCT03964792; EudraCT Number: 2018-001968-33) was designed by M.C. and sponsored by Assistance Publique–Hôpitaux de Paris. The study was conducted at Necker Children's Hospital in Paris. It was initiated in 2019 and enrolled a total of 6 patients, affected by SCD. All patients gave written informed consent. In accordance with French regulations, the patient 4 (minor at time of inclusion) has provided written assent, and her legal representative has given written consent. The authors collected and verified the completeness and accuracy of the data and the associated analyses, and vouch for adherence to the study protocol. The protocol was reviewed and approved by the French Comité de Protection des Personnes and relevant institutional ethics committees. The study included enhanced RBC transfusions at least three months before stem cell harvest. Moreover, doses of the myeloablative agent (busulfan) were adjusted on the basis of daily plasma pharmacokinetic analysis. Study protocol, statistical plan, eligibility criteria, primary and secondary outcomes, can be found at https://classic.clinicaltrials.gov/ct2/show/NCT03964792. The complete list of grade 3 adverse events AEs and severe adverse events SAEs is available in Supplementary Table 2.

Inclusion and exclusion criteria were:
Inclusion criteria:
- Age 12–20 years
- Diagnosis of HbSS or S-beta zero thalassemia by Hb electrophoresis or genetic analysis

- Clinical history or ongoing evidence of severe sickle cell anemia with one OR more of the following clinical complications demonstrating disease severity:

1. At least 3 vaso occlusive crises requiring hospitalization, under hydroxyurea or transfusion, within 2 years prior to enrollment
2. One severe ACS hospitalized in intensive care unit
3. At least 2 episodes of ACS within the prior 3 years), including one under HU.
4. Acute priapism (at least 2 episodes > 3 h in the preceding year or in the year prior to the start of a regular transfusion program), OR stuttering priapism ≥1 by week under sickle cell treatment (HU, transfusion or phlebotomy).
5. Cerebral vasculopathy confirmed by MRA (magnetic resonance angiography) without Moya-moya
6. Presence of sickle cell cardiomyopathy documented by Doppler echocardiography (left ventricular ejection fraction < 55% AND tricuspid regurgitation velocity >2.5 m/s on cardiac echocardiograph),
7. Tricuspid regurgitation velocity >2.8 m/s on cardiac echocardiograph without pulmonary hypertension confirmed by right heart catheterization (mPAP<25 mmHg)

- Failed hydroxyurea (HU) therapy, were unable to tolerate HU therapy, or, if 18 years of age or older, have actively made the choice to not take the recommended daily HU regimen. Inadequate clinical response to HU, defined as any one of the following outcomes, while on HU for at least 3 months: 2 or more acute sickle pain crises requiring hospitalization, no rise in Hb >1.5 g/dl from pre-HU baseline or requires transfusion to maintain Hb > 6.0 g/dL, Has an episode of ACS despite adequate supportive care measures.
- Karnovsky/Lansky performance score ≥ 60%
- Sexually active patients must be willing to use an acceptable method of double-barrier contraception for at least 12 months post-infusion (beyond 12 months at the discretion of the investigator) procedure for obtaining consent (adults, dependent minors, to give their consent, affiliation of a social security regime (or exemption).

     Exclusion criteria:
- Chromosomal (karyotyping) or molecular anomalies (detected by NGS) (i.e., 7 chromosomal monosomy)
- Existence of a matched sibling donor
- Patients who have started new treatment for SCD within 6 months of enrollment
- Hematologic evaluation: Leukopenia (WBC < 3000/μL) or neutropenia (ANC < 1000/μL) or thrombocytopenia (platelet count <100,000 /μL) (not due to an erythrapheresis procedure)
- PT/INR or PTT > 1.5 times upper limit of normal (ULN) or clinically significant bleeding disorder
- Evaluations within 6 months prior to screening visit:

1. ALT or AST > 3 times ULN
2. Liver Cirrhosis suspicion on echography, CT scan or MRI AND confirmed by histology
3. Cardiac evaluation: LVEF < 40% by cardiac echocardiogram or by MUGA scan or clinically significant ECG abnormalities
4. Stroke with significant CNS sequelae i.e., Rankin > 2
5. Lung interstitial infiltrate AND Forced Vital Capacity less than 70% AND DLCO less than 60% at steady state
6. Confirmed pulmonary hypertension defined by a right heart catheterization (PAPm>25 mmHg). Right heart catheterization is required if tricuspid regurgitation velocity >2.8 m/s on cardiac echocardiograph OR > 2.5 m/s with an abnormal Brain Natriuretic Peptide dosage or an important decrease in transcutaneous Hb O2 saturation during the 6 min' walk test.

- Seropositivity for HIV (Human Immunodeficiency Virus), HCV (Hepatitis C Virus), HTLV-1 (Human T-Lymphotropic Virus), or active Hepatitis B Virus, or active infection by CMV or parvovirus B19, based on positive blood PCR.
- Pregnancy or breastfeeding in a postpartum female
- Any current cancer or prior history of a malignant disease, with the exception of curatively treated non-melanoma skin cancer
- Immediate family member with an established or suspected Familial Cancer Syndrome
- Diagnosis of significant psychiatric disorder of the subject that could seriously impeded the ability to participate in the study
- Patients who failed previous HSCT and are severely ill
- Any clinically significant active infection
- Participation in another clinical study with an investigational drug within 30 days of screening
- Any condition, based on perspective of the medical monitor and treating investigator, which may lead to increased safety risk or inability to comply with the protocol

The primary and secondary endpoints to evaluate are listed in the clinical protocol (Supplementary Information File). All the additional analyses have been performed as exploratory.

*Patients* Conditioning and injection of genetically modified HSPCs were performed in the Pediatric Immunohematology Department or the Adult Hematology Department at Necker Children's Hospital (Paris, France). The drug product was manufactured in the Hospital's Cell and GT Laboratory at the Biotherapy Department. The follow-up included on-site regular patient visits in the Center of Clinical Investigations at Necker Enfants Malades Hospital and laboratory tests. Clinical and laboratory data were collected by the authors involved in the study. These included clinical status assessment, adverse event recording, immune cell hematological reconstitution, gene marking in cell subpopulations (VCN analyses) and hemoglobin quantification. Additional cell characterization assays were performed on an ad hoc basis.

For statistical analyses, for continuous variables, the median, minimum, and maximum values are usually given. For categorical variables, the proportion of treated patients in each category is presented. Specific statistical tests are described in figure legends and/or throughout the text.

*Healthy donors* Plerixafor-mobilized peripheral blood (MPB) or G-CSF and plerixafor-MPB samples from healthy donors were purchased by HemaCare (Northridge, CA, USA).

## Transduction protocol and VCN measurement
The lot of clinical-grade DREPAGLOBE lentiviral vector was manufactured at Yposkesi (Evry, France) under good manufacturing practices. One collection of plerixafor-mobilized CD34+ cells per patient was performed for both transduction and rescue. CD34+ cells were immunoselected using the CliniMACS system (Miltenyi Biotec). HSPCs were pre-activated for 48 h at a concentration of $10^6$ cells/mL in a pre-activation medium (PAM) composed of X-VIVO15 (Lonza) supplemented with recombinant human cytokines: 300 ng/mL SCF, 300 ng/mL Flt3-L and 100 ng/mL TPO (Cellgenix). After pre-activation, HSPCs ($3 \times 10^6$ cells/mL) were then transduced for 24 h in PAM supplemented with 10 μM PGE2 (Prostin E2, Pfizer), protamine sulfate (4 μg/mL, Protamine Choay, Sanofi) and Boost A (0.1 mg/ml, SirionBiotech). After transduction, cells were frozen until administration to the patient. Genomic DNA was extracted from an aliquot of HSPCs cultured for 14 days after transduction or from HSPC-derived BFU-E and CFU-GM and during the follow-up from sorted neutrophils, T cells, and on total PBMCs using a DNeasy Kit (Qiagen). The VCN was

determined in a quantitative PCR assay as previously described[21] or by digital droplet PCR as previously described[54].

## HSPC treatment with JAK inhibitors or TNF-α inhibitors

HSPCs were cultured in the PAM medium for 48 h in the presence of 0.5 μM ruxolitinib (Selleckchem), 200 nM baricitinib (TargetMol), 50 ng/μl infliximab (TargetMol) or 10 ng/ml etanercept (TargetMol). Forty-eight hours after treatment the transcriptomic profile was analyzed by bulk RNA-seq.

## Flow cytometry analysis of erythroid precursors and HSPCs

Flow cytometry analysis of bone marrow erythroid precursors was performed as previously described[20]. Patients' HSPCs were characterized using a multi labeled panel containing antibodies against the following markers: lineage (Lin) custom panel including anti-CD2, CD3, CD4, CD8, CD14, CD15, CD16, CD20, CD56, CD235a (Miltenyi Biotec, Bergisch Gladbach, Germany), CD34 clone 581 (Sony Biotechnologies, San Jose, CA, USA), CD133 clone 7 (Sony Biotechnologies), CD38 clone HB7 (Sony Biotechnologies), CD90 clone 5E10 (Sony Biotechnologies), CD45RA clone HI100 (Sony Biotechnologies), CD10 clone HI10a (BD Biosciences, Franklin Lakes, NJ, USA), CD110 clone BAH1 (BD Biosciences), CD71 clone CY1G4 (Sony Biotechnologies) and CD41 clone HIP8 (Sony Biotechnologies).

HSCs and progenitors were characterized using the following markers:

HSC: CD34+Lin-CD133+CD38-CD90+CD45RA-,
MPP: CD34+Lin-CD133+CD38-CD90-CD45RA-,
MLP: CD34+Lin-CD133+CD38-CD90-CD45RA+,
CMP: CD34+Lin-CD133-CD38+CD10-CD110-CD45RA-,
GMP: CD34+Lin-CD133-CD38+CD10-CD110-CD45RA+,
MEP: CD34+Lin-CD133-CD38+CD10-CD110+CD45RA-,
BNKP: CD34+Lin-CD133-CD38+CD10+.

Staining was analyzed with a Spectral ID7000 (Sony Biotechnologies) cell analyzer.

## Analysis of Hb and RBC properties

High-performance liquid chromatography (HPLC) analysis was performed using a NexeraX2 SIL-30AC chromatograph (Shimadzu) and the LC Solution software. Hemoglobin tetramers were separated by cation exchange (CE)-HPLC using a 2 cation-exchange column (Poly-CAT A, PolyLC, Columbia). Samples were eluted with a gradient mixture of solution A (20 mM bis Tris, 2 mM KCN, pH, 6.5) and solution B (20 mM bis Tris, 2 mM KCN, 250 mM NaCl, pH, 6.8). The absorbance was measured at 415 nm. Hemoglobin tetramers were quantified in BFU-E, total RBCs and in reticulocytes (CD71-positive cells sorted using anti-CD71 Microbeads; Miltenyi Biotec).

For the SCD study, results were compared to those from untransfused $\beta^S/\beta^S$ or $\beta^S/\beta^0$ patients selected from the ERYTHROPEDIE cohort monitored in our referral center (Henri Mondor Hospital, Creteil, France) and to healthy blood donors (HD) (from Etablissement Français du Sang). In accordance with the declaration of Helsinki, all patients gave their signed informed consent.

Intracellular Hb assessment by flow cytometry was performed as previously described[7,58]. Briefly, RBC samples frozen in glycerol were thawed, fixed and permeabilized before being incubated with mouse monoclonal antibodies (Ab) against human γ-globin chain (IQ Products: IQP-363-INT-4), $\beta^S$-globin chain (Rockland: 200-301-GS5) or $\beta^A$-globin chain (Rockland: 200-301-GS4). There is no cross reactivity among these 3 Abs. However, the anti-$\beta^A$-globin Ab also recognizes $\beta^{AS3}$-globin and δ-globin. Anti–globin Ab was coupled to phycoerythrin (R-PE) (IQ Products), whereas anti-$\beta^S$ and anti-$\beta^A$-globin Abs were coupled to Pacific Blue (PB) and to Alexa Fluor 647 (AF-647), respectively, using an Ab conjugation kit (Thermofisher). Reticulocytes were selected using an anti-CD71 (transferrin receptor) Ab coupled to PE (BD). Cells were incubated 15 min, shielded from light at room

temperature, then washed and analyzed on a BD LSR Fortessa flow cytometer (Becton Dickinson). The percentage of HbAS3+ cells was assessed using the gating strategy described in Supplementary Data 2. Briefly, after doublet exclusion, only HbS-positive cells were selected on a dot plot showing FSC-A vs. PB-A. A control of the day, made from a mix of RBCs from an untransfused SS individual and RBCs from an AS individual was used to set the gates "HbA-low" and "HbA-high" based in the AF-647 (anti-HbA antibody) fluorescence intensity. The gate "HbA-low" corresponds to SS RBCs for which the AF-647 fluorescence only comes from the intracellular HbA2. The gate "HbA-high" corresponds to AS RBCs, for which the AF-647 fluorescence comes from both HbA2 and HbA. The same gates with the same settings were then applied to RBC samples from GT patients acquired the same day. In P1, P2, P3, and P4, the "HbA-low" gate, which exhibits similar AF-647 mean fluorescence intensity (MFI) as SS-RBCs, corresponds to the RBC% without detectable intracellular HbAS3. In these patients, the gate "HbA-high", which exhibits higher AF-647 MFI than SS-RBC, corresponds to RBC containing intracellular HbA2 and detectable HbAS3, giving the percentage of HbAS3+ RBCs. A similar gating strategy was applied in HbS+CD71+ cells to assess the percentage of HbAS3-positive reticulocytes.

HbS level of polymerization was assessed by measuring the level of RBC sickling by microscopic observation after incubation for 20 min under 5% $O_2$ at 37 °C and RBC in vitro sickling was performed as previously described[5]. Briefly, RBC suspensions were added to a chambered coverslip, ibidi μ-Slide 8 Well (ibidi, Germany), and left to sediment for 30 min at 37 °C in a gas/temperature microscope stage incubator. RBC sickling was monitored by time-lapse microscopy at increasing hypoxia stages: 20, 10, 5, and 0% $O_2$. The duration of each stage was 30 min, pictures were taken at the end of each stage using an AxioObserver Z1 microscope. The percentage of sickled RBCs was determined in 4 different fields for each sample, using the following steps: total RBC count (T), sickled RBC count (S), calculated S/T ratio × 100.

HbS level of polymerization was assessed by in vitro Hb−$O_2$ dissociation and association curves on a Hemox analyzer (TCS scientific) to evaluate the $O_2$ saturation at various $PO_2$ pressures at pH = 7.4 and 37 °C.

RBC deformability was determined by laser diffraction analysis (ecktacytometry), using the Lorrca (Laser Optical Rotational Cell Analyzer−RR Mechatronics) by diluting blood in polyvinylpyrrolidone in normoxia under an increasing osmotic gradient (60 to 500 mOsm/kg) at shear stress set at 30 Pa and at 37 °C and under an oxygen gradient (by nitrogen-driven deoxygenation followed by air reoxygenation) at shear stress set at 30 Pa and at 37 °C following the manufacturer instructions.

The distribution of RBC density was performed using the phthalate method as described in ref. 5.

## Analysis of biological markers

Monitoring of biological markers was performed according to the local clinical laboratory tests and standards. Measure of plasma concentrations of hemoglobin, heme and hemopexin was done by spectrophotometry. Quantification of iron content in tissues was performed by 1.5 Tesla MRI in liver (R2*/T2* relaxometry) and heart (T2* relaxometry).

## Integration site analysis

IS analysis was performed on DNA from whole blood samples using ligation-mediated PCR and analyzed as described[59,60]. Relative clone size was quantified using the Sonic Abundance method[61].

## NGS sequencing of PCR amplicons

The following genes were PCR-amplified and subjected to NGS sequencing:

*AKT1, ALK, ARID1A, ASXL1, ATM, B2M, BCL11B, BCOR, CARD11, CBL, CBLB, CCND3, CCR4, CCR7, CD28, CD58, CDKN2A, CEBPA, CISH, CNOT3, CREBBP, CSNK1A1, CSNK2A1, CSNK2B, CTCF, CTNNB1, CXCR4, DDX3X, DNM2, DNMT3A, EED, EP300, ETV6, EZH2, FAS, FBXW7, FLT3, FOXO3, FPGS, FYN, GATA3, GPR183, HLA-B, HNRNPA2B1, HRAS, IDH1, IDH2, IKZF1, IL7R, IRF2BP2, IRF4, JAK1, JAK2, JAK3, JAKMIP2, KDM6A, KIT, KMT2A, KMT2C, KMT2D, KRAS, LCK, LEF1, LMO1, LMO2, LRP1B, MGA, MSH2, MSH6, MYB, MYCN, NF1, NOTCH1, NR3C1, NR3C2, NRARP, NRAS, NT5C2, PHF6, PIAS1, PIK3CA, PIK3CD, PIK3R1, PLCG1, PMS2, POT1, PRKCB, PRPS1, PRPS2, PTEN, PTPN11, PTPN2, PTPN6, PTPRC, PTPRD, RB1, RELN, RHOA, RPL10, RPL5, RUNX1, SETD2, SF3B1, SH2B3, SKP2, SLC19A1, SLC7A11, SMARCA4, SOCS1, SOCS3, SRSF2, STAT1, STAT3, STATSA, STATSB, SUZ12, TAU, TBL1XR1, TDRD6, TENM2, TET1,TET2, TET3, TP53, TRDD1, TRDD2, TRDD3, TRDJ1, TRDJ2,TRDJ3, TRDJ4, TSR1, TYK2, U2AF1, USP7, USP9X, VAV1, WT1, ZFHX4, ZFP36L2, ZRSR2*

## Bulk RNA-seq analysis

RNA was isolated using the RNeasy Micro Kit (Qiagen) with a DNase step. RNA integrity and concentration were assessed using capillary electrophoresis and the Fragment Analyzer (Agilent). RNA-seq libraries were prepared from 100 ng of total RNA, using the Universal Plus mRNA (Nugen-Tecan). The amplified cDNA was sequenced on a NovaSeq6000 system (Illumina) obtaining ~50 million reads per library.

The raw read counts were normalized with the DESeq2 package, based on the library size and testing for differential expression between conditions[62]. Coding genes were extracted from gencodeV30. Next, the noise filter was used to retain only genes that had at least one sample with an expression value greater than 20 counts before the pathway enrichment analysis. Genesets enrichment were investigated with MSigDB, using a hypergeometric test on a pre-filter dataset ($p < 0.05$ and fold-change (FC) $> 1.2$ or $< -1/1.2$). The output false discovery rate had to be below 0.05.

Representation and quantification of module activity (ROMA) was applied to DEGs in HSPCs. ROMA calculates a module score for a set of samples and is based on the simplest single-factor linear model of gene regulation whose first principal component approximates the expression data[23].

## scRNA-seq analysis

Empty droplets were excluded with the DropletUtils package, with an FDR threshold of 0.01. Cells with more than 15% of mitochondrial genes and less than 3000 unique molecular identifiers (UMI) were removed. We analyzed only cells expressing CD34 to remove contaminants. Doublets were removed using the DoubletFinder package. As HSPCs differ in their maturity (translating into difference in expression abundance from one cell to another), the expression matrix for each sample was normalized using deconvolution rather than standard library size methods[63]. The gene expression was then restricted to protein-encoding genes. The 6000 highly variable genes were found with the Seurat FindVariableFeatures function and its default parameters.

We then used Cell-ID as a robust statistical method for gene signature extraction and cell identity recognition on the basis of single-cell RNA-seq data[64]. It incorporates a multiple correspondence analysis and simultaneously represents cells and genes in low-dimension space. The genes are then ranked by their Euclidean distance from each individual cell, which provides unbiased per-cell gene signatures. For each group (SCD, HD), we annotated each cell based on their top 400 most specific genes using CellID and published signatures[24]. The Cell-ID method defines the gene ranking in each cell in the dataset (54,689 cells in total), evaluates whether a cell accurately matches a particular reference signature, and determines the cell's identity (Cell ID) based on the top $p$-value ($p < 0.05$). The enrichment score is based on the $-\log_{10}(p$-value).

In order to identify the DEGs in HSC subpopulation, we used the MAST approach[65] (https://github.com/RGLab/MAST) based on statistical models tailored to single-cell data, allowing inference for genes with sparse expression. These models can handle a more complex variance structure, such as expected correlations between cells derived from the same individual. DEGs were identified with a Hurdle model (implemented with MAST v1.16.0) testing the 6000 highly variable genes in the dataset, and adjusting with a cellular detection rate parameter that corresponds to the number of genes detected in a cell.

To analyze the enrichment pathway, we applied a hypergeometric test for pathway enrichment using Hallmark geneset from MSigDB database or Custom geneset (Supplementary Data 6).

For more details on the transcriptomic analyses were performed as previously described[24].

## HSPC xenotransplantation in NSG mice

Non-obese diabetic severe combined immunodeficiency gamma (NSG) mice (NOD.CgPrkdcscid Il2rgtm1Wjl/SzJ, Charles River Laboratories, St Germain sur l'Arbresle, France) were housed in a specific pathogen-free facility. Mice at 6 to 8-weeks of age were conditioned with busulfan (Sigma, St Louis, MO, US) injected intraperitoneally (15 mg/kg body weight/day) 24 h, 48 h, and 72 h before transplantation. Mock- or LV-transduced CD34+ cells (500,000 cells/mouse) were transplanted into NSG mice via retro-orbital sinus injection. Neomycin and acid water were added in the water bottle. At 18 to 19 weeks post-transplantation, NSG recipients were sacrificed. Cells were harvested from femur BM and stained with antibodies against murine or human surface markers such as murine CD45 REA737 (Miltenyi Biotec, Bergisch Gladbach, Germany); human CD45 REA747 (Miltenyi Biotec); human CD3 REA613 (Miltenyi Biotec); human CD14 clone MφP9 (BD Biosciences, Franklin Lakes, NJ, USA); human CD15 80H5 (Beckman Coulter, Brea, CA, USA); human CD19 clone HIB19 (Sony Biotechnologies); human CD36 clone CB38 (BD Biosciences); human CD71 clone M-A712 (BD Biosciences); human CD34 clone 4H11 (eBioscience, Invitrogen Carlsbad, California, United States). Stained cells were analyzed by flow cytometry using a MACSQuant analyzer (Miltenyi Biotec) and the FlowJo software. Human bone marrow CD45+ cells were sorted by immunomagnetic selection with AutoMACS (Miltenyi Biotec) after immunostaining with the CD45 MicroBead Kit (Miltenyi Biotec). DNA was extracted and VCN was determined as described above. All experiments and procedures were performed in compliance with the French Ministry of Agriculture's regulations on animal experiments and were approved by the regional Animal Care and Use Committee (APAFIS#2101-2015090411495178 v4). All animal experiments were performed according to the ARRIVE guidelines.

## Statistics

Specific statistical tests are described in figure legends and/or throughout the text. The statistical plan can be found at https://clinicaltrials.gov/ct2/show/NCT02151526.

## Transfer vector sequence

The sequence is available in the Supplementary Information File.

## Reporting summary

Further information on research design is available in the Nature Portfolio Reporting Summary linked to this article.

## Data availability

Bulk RNAseq and Single Cell RNAseq data are available at Biostudies EMBL-EBI (S-BSST1524, S-BSST1525, S-BSST1526, S-BSST1257, and S-BSST1258). Lentiviral integration sites sequencing data are available under NCBI BioProject Accession ID PRJNA1225189. Source data are provided with this paper.

## Code availability

The full code listing used for the analysis and the figures are available online in the following repository: https://doi.org/10.5281/zenodo.10069444.

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

## Acknowledgements

This work was supported by State funding from the Agence Nationale de la Recherche under "Investissements d'avenir" program (ANR-10-IAHU-01: funding to the Imagine Institute), by the Paris Ile de France Region under "DIM Thérapie génique" initiative (funding to Annarita Miccio's and Marina Cavazzana's labs), by the Dior chair for tailored medicine (funding to Marina Cavazzana's lab), by the European Union's Horizon Europe research and innovation program (under grant agreement # 101057659; funding to Annarita Miccio's lab), by the AFM-TELETHON (grant #23879; funding to Annarita Miccio's lab) and by the Etablissement Français du Sang (funding to Pablo Bartolucci's lab). We thank the patients and their families for their cooperation during the study. We also thank Jinmi Baek and Jouda Marouene at the Centre d'Investigation Clinique - Unité de Recherche Clinique (Hôpital Universitaire Necker Enfants-Malades), Marine Luka at the Imagine Institute's single-cell facility, Christine Bole and colleagues at the Imagine Institute's genomics facility, Nicolas Cagnard and Cecile Masson at the Imagine Institute's bioinformatics facility.

## Author contributions

Mari.C. was the principal investigator of the trial and P.B. was the study director. S.S., E.M., A.Ch., N.H., C.R., C.A., O.L., A.D., M.B., T.F., J.S.D., A.P., J.E.E., A.G., E.T., F.M., Al.M., V.A., and W.E.N., P.B., E.S., M.S., and An.M. performed biological assays and/or analyzed and interpreted data. E.M, C.R., C.A., and O.L. performed vector transduction. S.S., F.B., A.D., and E.S. performed bioinformatic analyses. L.J., S.M., J.M.T., P.B., M.S., and Mari.C. were involved in patients' inclusion and provided clinical follow up. Am.M., Mart.C., B.N., O.H., A.Co., S.R., performed transplantation and/or clinical services during hospitalization. S.S., E.S., M.S., An.M., and Mari.C wrote primarily the manuscript and all authors critically revised the manuscript.

## Competing interests

The authors declare no competing interests.

## Additional information

Steicy Sobrino[1,2], Laure Joseph[3], Elisa Magrin[3,4], Anne Chalumeau [1], Nicolas Hebert [5,6], Alice Corsia[3,7], Adeline Denis[2], Cécile Roudaut[3,4], Clotilde Aussel[3,4], Olivia Leblanc[3,4], Mégane Brusson [1], Tristan Felix [1], Jean-Sebastien Diana[3,4], Angelina Petrichenko [8], Jana El Etri[1,2], Auria Godard[9], Eden Tibi[3], Sandra Manceau[3,7], Jean Marc Treluyer[10,11], Fulvio Mavilio [12], Frederic D. Bushman [8], Ambroise Marcais [3,7], Martin Castelle[13], Benedicte Neven[13], Olivier Hermine[3,7], Sylvain Renolleau [3,7], Alessandra Magnani[3,4], Vahid Asnafi [14], Wassim El Nemer[9], Pablo Bartolucci[5,15], Emmanuelle Six [2,17], Michaela Semeraro[10,16,17], Annarita Miccio [1,17] & Marina Cavazzana [3,4,17] ✉

[1]Laboratory of Chromatin and Gene Regulation During Development, University Paris Cite, UMR1163 INSERM, Imagine Institute, Paris, France. [2]Laboratory of Human Lymphohematopoiesis, INSERM, Imagine Institute, Paris, France. [3]Departement of Biotherapy Necker Children's Hospital, Assistance Publique-Hôpitaux de Paris (AP-HP) Groupe Hospitalier Universitaire Centre, Université Paris Cité, Paris, France. [4]Biotherapy Clinical Investigation Center, AP-HP, INSERM, Institut Imagine, Paris, France. [5]Univ Paris Est Créteil, IMRB, Laboratory of Excellence LABEX GRex, Créteil, France. [6]Etablissement Français du Sang, Créteil, France. [7]Department of Adult Hematology, Necker Hospital, Assistance Publique-Hôpitaux de Paris, Laboratory of Excellence LABEX GRex, Paris, France. [8]Department of Microbiology, Perelman School of Medicine, University of Pennsylvania, Philadelphia, PA, USA. [9]Aix Marseille Univ, CNRS, EFS, ADES, Labex GR-Ex, Marseille, France. [10]Centre d'Investigation Clinique-Unité de Recherche Clinique, Hôpital Universitaire Necker Enfants-Malades, GH Paris Centre, Paris, France. [11]Université Paris Cité, Paris, France. [12]Department of Life Sciences, University of Modena and Reggio Emilia, Modena, Italy. [13]Pediatric Immunology and Hematology Department, Hôpital Necker Enfants-Malades, Paris, France. [14]Laboratory of Onco-Hematology, Necker Children's Hospital, Assistance Publique-Hôpitaux de Paris (AP-HP), Paris, France. [15]Paris-East Créteil University, Henri Mondor University Hospitals, APHP, Sickle Cell Referral Center-UMGGR, Créteil, France. [16]Université Paris Cité, Inserm, Pharmacologie et évaluations des thérapeutiques chez l'enfant et la femme enceinte, Paris, France. [17]These authors jointly supervised this work: Emmanuelle Six, Michaela Semeraro, Annarita Miccio, Marina Cavazzana. ✉e-mail: m.cavazzana@aphp.fr

