## [Transparent Peer Review file · Nature Communications]

Severe inflammation and lineage skewing are associated with poor engraftment of engineered hematopoietic stem cells in patients with sickle cell disease

Corresponding Author: Professor Marina Cavazzana

Version 0:

Reviewer comments:

Reviewer #3

(Remarks to the Author)

I found this manuscript to be very challenging and am not sure exactly how to interpret the findings.

Specific concerns:

1. The study used a different vector than previously successful vectors.
2. The outcomes were clearly unsatisfactory--even the "more successful" patients had ongoing problems that led the study to be closed after only 4 patients.
3. The findings regarding inflammation are interesting, but are the generalizable? Patient 3 had high mega precursors--interesting, how does this help us? Patient 4 had high TNF and interferon scores in HSC's (but some of this occurred in P2 as well, so how does this help the inflammation argument? Pt 2 had the highest VCN).
4. The inflammatory signature in the thalassemia patient with low engraftment is interesting. The treatment of cells from patients P3 and P4 with anti-inflammatory agents leading to less inflammation is expected--the key question is whether this would make a difference in patient outcome.

The big question I have is whether these issues can be explained by this being a poor vector for gene therapy or that there were less than optimal approaches in obtaining and processing the gene product. What convincing arguments can you make to say that your findings are generalizable? Have you used other more successful vectors and been able to replicate your findings? (can't do another gene therapy trial, but what about in vitro studies?). Were your patients sicker than patients on other trials? Unsure about what the next step is other than use a different vector, but is there sufficient evidence for us to use ruxilitinib in processing?

(Remarks on code availability)

seems fine

Reviewer #4

(Remarks to the Author)

I would like to thank the authors for responding to the reviewers' comments and including the most relevant ones in the manuscript.

As mentioned before, I would also like to thank the authors for designing such a relevant study for the future improvement of gene therapy approaches in SCD and probably in other diseases as well.

(Remarks on code availability)

Point-by-point-reply to reviewers' comments

Reviewer #3 (Remarks to the Author):

1. The study used a different vector than previously successful vectors.

In this study, we have used two different vectors, namely (i) DREPAGLOBE, a vector adapted for use in SCD gene therapy (by introducing three polymerization-inhibiting amino acid substitutions into the β -globin chain), from the GLOBE vector that was previously successfully used in 9 patients with beta-thalassemia^{1,2} and (ii) BB305, a vector successfully used in clinical trials for both beta-thalassemia and sickle cell disease³⁻⁷. We have clarified this point in the results on page 8 line 362.

Of note, as mentioned at the beginning of the discussion on page 9 lines 399-402 "In contrast to our starting hypothesis (i.e. that the introduction of two additional anti-sickling amino acids into the DREPAGLOBE vector would give better results), the clinical and biological outcomes were similar to those observed with the BB305 vector in patients with comparable VCNs."

However, we agree with the reviewer that optimizing the transduction protocol and LV design could also improve the outcome of SCD GT. To ameliorate the outcome of the future clinical trial, we will employ an optimized protocol based on the use of effective transduction enhancers^{3,7,8} and an optimized vector that corrects the SCD phenotype with greater potency and hence lower VCN⁹.

We have now clarified these points in the discussion on page 11 lines 517-521.

2. The outcomes were clearly unsatisfactory--even the "more successful" patients had ongoing problems that led the study to be closed after only 4 patients.

In this study, we have shown that gene therapy efficacy might depend on the number of infused HSCs and intrinsic, engraftment-impairing inflammatory alterations in HSCs.

While the clinical and biological outcomes were similar to those observed with the BB305 vector in SCD patients with comparable VCN, we acknowledge the fact that we achieved an overall low VCN, which, as demonstrated for the BB305 vector, is not sufficient to fully correct the SCD clinical phenotype. However, it is worth noting that for P1 and P2 gene therapy has been transformative: they are no longer transfusion-dependent, experienced only occasional grade 3 VOCs post-GT (mainly triggered by specific factors, such as cold seawater bath) with shorter hospitalizations and significantly reduced analgesic treatment compared to pre-GT period. Finally, for P1 and P2, we have also observed a positive impact of GT on their quality of life and well-being (see paragraph page 5 and lines 212-235).

To ameliorate the outcome of the future clinical trial that we will conduct for SCD patients, we will employ an optimized protocol based on: (i) the treatment of the patients with anti-inflammatory drugs (to collect high-quality HSCs); (ii) the use of a high infused HSC number; (iii) the use of better transduction enhancers currently used in BB305-base clinical trials for SCD^{3,7,8}, (iv) an optimized vector that corrects the SCD phenotype with greater potency and hence lower VCN⁹.

We have now clarified these points in the discussion on pages 10-11 lines 479-485 and 517-521.

3. The findings regarding inflammation are interesting, but are the generalizable?

Patient 3 had high mega precursors--interesting, how does this help us?

Patient 4 had high TNF and interferon scores in HSC's (but some of this occurred in P2 as well, so how does this help the inflammation argument? Pt 2 had the highest VCN).

In patient 3, not only we observed an increase in megakaryocytic precursors but also a 3-fold increase in megakaryocyte-biased HSCs (**Fig. 4a**) correlated with a significant increase in IL1B expression in this population and an aberrant expression of the CD71 marker on the most immature HSC as shown by flow cytometry analysis (**Extended data Fig. 5**). The CD71+ HSC population has been previously characterized by others to show no self-renewal activity¹⁰

and therefore could explain the loss of engraftment capacity in patient 3. We have now clarified this point in the discussion on page 10 lines 445-449.

Similarly to Patient 2, Patient 4 HSPCs showed a myeloid bias (**Fig. 3h, Extended Data Fig. 5a-c and Fig. 4g**). However, despite this similarity, Patient 4 presented with a much higher inflammation score compared with P2 (**Fig. 4e and f**), suggesting chronic accumulation of inflammatory stress that could explain the loss of engraftment capacity in Patient 4. We have now clarified this point in the discussion on page 10 lines 459-462.

In order to further understand the consequences of the various inflammation levels observed in SCD P3 and P4, and in TDT P3, and if our findings can be generalizable, we took advantage of the recently published human aging signature¹¹ and applied it on our datasets. The Aged vs Young HSC enrichment score was the highest for SCD P3, P4 and TDT P3 in HSCs and HSC-enriched populations (Novel Figure **Extended Data Fig. 8a-d**). As expected for HDs, the Aged vs Young HSC median score in HSCs was positively correlated with the age ($r=0.89$; $p=0.045$). When combining HD with patients, we lost the correlation ($r=0.38$, not significant), as SCD P3 and P4, and TDT P3 presented with abnormally high Aged vs Young HSC score (Novel Figure **Extended Data Fig. 8e**). Altogether, these results confirm that chronic inflammation can lead to premature aging of HSCs, which could affect their engraftment capacity. Therefore, we believe that our findings can be generalizable and allow us to potentially pre-screen the patients for ensuring the success of gene therapy.

We added these new analyses in the results on pages 8-9 lines 372-379 and in the novel figure **Extended Data Fig. 8**.

4. The inflammatory signature in the thalassemia patient with low engraftment is interesting. The treatment of cells from patients P3 and P4 with anti-inflammatory agents leading to less inflammation is expected--the key question is whether this would make a difference in patient outcome.

We agree with the reviewer and we will address this point in future clinical trials, as mentioned in the discussion on pages 10-11 lines 481-486.

The big question I have is whether these issues can be explained by this being a poor vector for gene therapy or that there were less than optimal approaches in obtaining and processing the gene product.

While we have used two vectors that were successfully used in gene therapy trials for beta-hemoglobinopathies, we agree with the reviewer that optimizing the transduction protocol and LV design could also improve the outcome of SCD GT. Please see answers to comments 1 and 2.

What convincing arguments can you make to say that your findings are generalizable?

Please See answer to comment 3.

Have you used other more successful vectors and been able to replicate your findings? (can't do another gene therapy trial, but what about in vitro studies?).

Please see answers to comments 1 and 2.

Were your patients sicker than patients on other trials?

The clinical phenotype of these patients was not different from that of individuals treated in other trials.

Unsure about what the next step is other than use a different vector, but is there sufficient evidence for us to use ruxolitinib in processing?

Treatment with ruxolitinib might have the potential to improve HSC fitness. Interestingly, two studies recently showed that ruxolitinib ex vivo treatment improves the self-renewal capacity of HSCs and enhances their maintenance in culture^{12,13}. We are currently evaluating a variety

of anti-inflammatory drugs that could enhance HSC fitness (beyond JAK inhibitors). For example, we have tested TNF- α inhibitors, which also reduced the inflammation signature in P3 and P4. We have now included these data in the novel panels in **Fig. 5d, g and h**. To validate the use of these drugs to improve SCD GT, further in vivo studies evaluating the impact on SCD HSCs and the safety profile are required and are ongoing in our lab. We have now clarified this point in the discussion on pages 9-10 lines 381-394, 479-481 and 484-485.

Reviewer #3 (Remarks on code availability):

seems fine

Thank you.

Reviewer #4 (Remarks to the Author):

I would like to thank the authors for responding to the reviewers' comments and including the most relevant ones in the manuscript.

As mentioned before, I would also like to thank the authors for designing such a relevant study for the future improvement of gene therapy approaches in SCD and probably in other diseases as well.

We thank Reviewer 4 for his comments.

References

1. Miccio, A. *et al.* In vivo selection of genetically modified erythroblastic progenitors leads to long-term correction of beta-thalassemia. *Proc Natl Acad Sci U S A* **105**, 10547–10552 (2008).
2. Markt, S. *et al.* Intrabone hematopoietic stem cell gene therapy for adult and pediatric patients affected by transfusion-dependent β -thalassemia. *Nat Med* **25**, 234–241 (2019).
3. Kanter, J. *et al.* Biologic and Clinical Efficacy of LentiGlobin for Sickle Cell Disease. *N Engl J Med* **386**, 617–628 (2022).
4. Ribeil, J.-A. *et al.* Gene Therapy in a Patient with Sickle Cell Disease. *N Engl J Med* **376**, 848–855 (2017).
5. Magrin, E. *et al.* Long-term outcomes of lentiviral gene therapy for the β -hemoglobinopathies: the HGB-205 trial. *Nat Med* **28**, 81–88 (2022).
6. Thompson, A. A. *et al.* Gene Therapy in Patients with Transfusion-Dependent β -Thalassemia. *N Engl J Med* **378**, 1479–1493 (2018).
7. Locatelli, F. *et al.* Betibeglogene Autotemcel Gene Therapy for Non- β^0/β^0 Genotype β -Thalassemia. *N Engl J Med* **386**, 415–427 (2022).
8. Masiuk, K. E. *et al.* PGE2 and Poloxamer Synperonic F108 Enhance Transduction of Human HSPCs with a β -Globin Lentiviral Vector. *Mol Ther Methods Clin Dev* **13**, 390–398 (2019).
9. Brusson, M. *et al.* Novel lentiviral vectors for gene therapy of sickle cell disease combining gene addition and gene silencing strategies. *Mol Ther Nucleic Acids* **32**, 229–246 (2023).
10. Mende, N. *et al.* Unique molecular and functional features of extramedullary hematopoietic stem and progenitor cell reservoirs in humans. *Blood* **139**, 3387–3401 (2022).
11. Jakobsen, N. A. *et al.* Selective advantage of mutant stem cells in human clonal hematopoiesis is associated with attenuated response to inflammation and aging. *Cell Stem Cell* **31**, 1127-1144.e17 (2024).
12. Williams, M. J. *et al.* Maintenance of hematopoietic stem cells by tyrosine-unphosphorylated STAT5 and JAK inhibition. *Blood Adv* (2024) doi:10.1182/BLOODADVANCES.2024014046.
13. Johnson, C. S. *et al.* Adaptation to ex vivo culture reduces human hematopoietic stem cell activity independently of the cell cycle. *Blood* **144**, 729–741 (2024).

Point-by-point-reply to reviewers' comments

We thank the reviewers for the careful revision.